# AC-PKAN: Attention-Enhanced and Chebyshev Polynomial-Based Physics-Informed Kolmogorov–Arnold Networks

## Abstract

This paper introduces AC-PKAN, an advanced framework for Physics-Informed Neural Networks (PINNs) that integrates Kolmogorov–Arnold Networks (KANs) with Chebyshev Type-I polynomials and incorporates both internal and external attention mechanisms. Traditional PINNs based on Multilayer Perceptrons (MLPs) encounter challenges when handling complex partial differential equations (PDEs) due to vanishing gradients, limited interpretability, and computational inefficiency. To address these issues, we enhance the model from both external and internal perspectives. Externally, we propose a novel Residual Gradient Attention (RGA) mechanism that dynamically adjusts loss term weights based on gradient norms and residuals, thereby mitigating gradient stiffness and residual imbalance. Internally, AC-PKAN employs point-wise Chebyshev polynomial-based KANs, wavelet-activated MLPs with learnable parameters, and internal attention mechanisms. These integrated components improve both training efficiency and prediction accuracy. We provide mathematical proofs demonstrating that AC-PKAN can theoretically solve any finite-order PDE. Experimental results from five benchmark tasks across three domains show that AC-PKAN consistently outperforms or matches state-of-the-art models such as PINNsFormer, establishing it as a highly effective tool for solving complex real-world engineering problems.

## 1 Introduction

Numerical solutions of partial differential equations (PDEs) are essential in science and engineering (Zienkiewicz & Taylor, 2005; Liu, 2009; Fornberg, 1998; Brebbia et al., 2012). Physics-informed neural networks (PINNs) (Lagaris et al., 1998; Raissi et al., 2019) have emerged as a promising approach in scientific machine learning. Traditional PINNs typically employ multilayer perceptrons (MLPs) (Cybenko, 1989) due to their ability to approximate nonlinear functions (Hornik et al., 1989) and their success in various PDE-solving applications (Yu et al., 2018; Han et al., 2018).

However, PINNs encounter limitations, including difficulties with multi-scale phenomena (Kharazmi et al., 2021), the curse of dimensionality in high-dimensional spaces (Jagtap & Karniadakis, 2020), and challenges with nonlinear PDEs (Yuan et al., 2022). These issues arise from both the complexity of PDEs and limitations in PINN architectures and training methods.

To address these challenges, existing methods focus on improving either the internal architecture of PINNs or their external learning strategies. Internal improvements include novel architectures like Quadratic Residual Networks (Qres) (Bu & Karpatne, 2021), First-Layer Sine (FLS) (Wong et al., 2022), and PINNsformer (Zhao et al., 2023). External strategies are discussed in detail in Section 2. Nevertheless, traditional PINNs based on MLPs still suffer from issues like lack of interpretability (Cranmer, 2023), overfitting, vanishing or exploding gradients, and scalability problems (Bachmann et al., 2024). As an alternative, Kolmogorov–Arnold Networks (KANs) (Liu et al., 2024b), inspired by the Kolmogorov–Arnold representation theorem (Kolmogorov, 1961; Braun & Griebel, 2009), have been proposed to offer greater accuracy and interpretability. KANs can be viewed as a combination of Kolmogorov networks and MLPs with learnable activation functions (Köppen, 2002; Sprecher & Draghici, 2002). Various KAN variants have emerged by replacing the B-spline func-

tions (SS, 2024; Bozorgasl & Chen, 2024; Xu et al., 2024). Although they still face challenges (Yu et al., 2024), KANs have shown promise in addressing issues like interpretability (Liu et al., 2024a) and catastrophic forgetting (Vaca-Rubio et al., 2024) in learning tasks (Samadi et al., 2024). Recent architectures like KINN (Wang et al., 2024b) and DeepOKAN (Abueidda et al., 2024) have applied KANs to PDE solving with promising results.

Despite the potential of KANs, the original KAN suffers from high memory consumption and long training times due to the use of B-spline functions (Shukla et al., 2024). To address these limitations, we propose the Attention-Enhanced and Chebyshev Polynomial-Based Physics-Informed Kolmogorov–Arnold Networks (AC-PKAN). Our approach replaces B-spline functions with first-kind Chebyshev polynomials, forming the Cheby1KAN layer (SynodicMonth, 2024), eliminating the need for grid storage and updates. By integrating Cheby1KAN with linear layers and incorporating internal attention mechanisms based on input features, AC-PKAN efficiently models complex nonlinear functions and focuses on different aspects of input features at each layer. Additionally, we introduce an external attention mechanism that adjusts loss weights dynamically based on gradient information and point-wise residuals, alleviating issues such as residual imbalance and gradient flow stiffness, and enhancing training stability and efficiency. To our knowledge, AC-PKAN is the first PINN framework to integrate internal and external attention mechanisms into KAN layers, effectively addressing many issues of original KANs and PINNs. Our key contributions can be summarized as follows:

- **Novel Loss Weighting Mechanism:** We propose the Residual Gradient Attention (RGA) mechanism, which dynamically adjusts loss term weights based on gradient norms and residual magnitudes, alleviating gradient stiffness and residual imbalance, thereby enhancing convergence and predictive performance.

- **Innovative Framework:** We develop AC-PKAN by integrating Cheby1KAN layers with attention-enhanced MLPs and a new learnable activation function, *Wavelet*. We prove that our framework can approximate and solve arbitrary finite-order PDEs, significantly improving performance on complex PDE problems and offering flexible activation function learning, improved parameter efficiency, and superior generalization compared to traditional PINNs.

- **Extensive Experiments:** We evaluate AC-PKAN on three categories of five benchmark tasks involving 12 models, demonstrating that it achieves best or near-best performance. We also analyze loss landscapes and conduct ablation studies to show the impact of our proposed modules, addressing a research gap in KAN research.

## 2 RELATED WORK

**External Learning Strategies for PINNs.** Various external strategies have been proposed to address the limitations of PINNs. Loss weighting methods, such as PINN-LRA (Wang et al., 2021), PINN-NTK (Wang et al., 2022), and PINN-RBA (Anagnostopoulos et al., 2024), rebalance loss terms using gradient norms, neural tangent kernels, and residual information to enhance training efficiency. Optimizer improvements like MultiAdam (Yao et al., 2023) aid convergence during multi-scale training. Advanced sampling strategies, including AAS (Tang et al., 2023), which combines optimal transport theory with adversarial methods, RoPINN (Pan et al., 2024), which utilizes Monte Carlo sampling for regional optimization, RAR (Wu et al., 2023), which applies residual-driven resampling, and PINNACLE (Lau et al., 2024), which adaptively co-optimizes the selection of all types of training points, have been developed to improve performance. Enhanced loss functions like gPINN (Yu et al., 2022) and vPINN (Kharazmi et al., 2019) incorporate gradient enhancement and variational forms, respectively. Adaptive activation functions in LAAF (Jagtap et al., 2020a) and GAAF (Jagtap et al., 2020b) accelerate convergence and handle complex geometries. Domain decomposition methods such as FBPINN (Moseley et al., 2023) and hp-VPINN (Kharazmi et al., 2021) train subnetworks on subdomains and use higher-order polynomial projections for refinement.

**Variants of KAN.** Since the introduction of KAN (Liu et al., 2024b), various variants have been developed to enhance performance and computational efficiency by modifying the basis functions. FastKAN (Li, 2024) replaces third-order B-spline bases with radial basis functions (RBFs) for accelerated computation. Chebyshev1KAN (SynodicMonth, 2024) and Chebyshev2KAN (SS, 2024) use

first and second kinds of Chebyshev polynomials, leveraging their advantageous properties. Rational KAN (rKAN) (Afzalaghaei, 2024b) and Fractional KAN (fKAN) (Afzalaghaei, 2024a) incorporate trainable adaptive rational-orthogonal and fractional-orthogonal Jacobi functions, enhancing adaptability and approximation capabilities. FourierKAN (GistNoesis, 2024) replaces spline coefficients with one-dimensional Fourier coefficients, serving as a substitute for linear layers and non-linear activation functions. Preliminary comparisons (Jerry-Master, 2024) indicate that Cheby1KAN currently offers the best efficiency and performance.

# 3 METHODOLOGY

**Preliminaries:** Let $\Omega \subset \mathbb{R}^d$ be an open set with boundary $\partial \Omega$. Consider the PDE:

$$
\begin{aligned}
\mathcal{D}[u(\boldsymbol{x},t)] &= f(\boldsymbol{x},t), \quad (\boldsymbol{x},t) \in \Omega, \\
\mathcal{B}[u(\boldsymbol{x},t)] &= g(\boldsymbol{x},t), \quad (\boldsymbol{x},t) \in \partial\Omega,
\end{aligned}
\tag{1}
$$

where $u$ is the solution, $\mathcal{D}$ is a differential operator, and $\mathcal{B}$ represents boundary or initial conditions. Let $\hat{u}$ be a neural network approximation of $u$. PINNs minimize the loss:

$$
\mathcal{L}_{\text{PINNs}} = \lambda_r \sum_{i=1}^{N_r} \|\mathcal{D}[\hat{u}(\boldsymbol{x}_i,t_i)] - f(\boldsymbol{x}_i,t_i)\|^2 + \lambda_b \sum_{i=1}^{N_b} \|\mathcal{B}[\hat{u}(\boldsymbol{x}_i,t_i)] - g(\boldsymbol{x}_i,t_i)\|^2,
\tag{2}
$$

where $\{(\boldsymbol{x}_i,t_i)\} \subset \Omega$ are residual points, $\{(\boldsymbol{x}_i,t_i)\} \subset \partial\Omega$ are boundary or initial points, and $\lambda_r$, $\lambda_b$ balance the loss terms. The goal is to train $\hat{u}$ to minimize $\mathcal{L}_{\text{PINNs}}$ using machine learning techniques.

## 3.1 RESIDUAL-AND-GRADIENT BASED ATTENTION

In the standard PINNs framework, the total loss $\mathcal{L}_{\text{PINNs}}$ comprises the residual loss $\mathcal{L}_r$, boundary condition loss $\mathcal{L}_{bc}$, and initial condition loss $\mathcal{L}_{ic}$:

$$
\mathcal{L}_{\text{PINNs}} = \lambda_r \mathcal{L}_r + \lambda_{bc} \mathcal{L}_{bc} + \lambda_{ic} \mathcal{L}_{ic},
\tag{3}
$$

where $\lambda_r$, $\lambda_{bc}$, and $\lambda_{ic}$ are weighting coefficients.

To improve training efficiency and accuracy, we propose a novel **Residual-and-Gradient Based Attention (RGA)** mechanism that adaptively reweights loss components by considering both residual magnitudes and gradient norms. This approach ensures balanced and efficient optimization, particularly addressing challenges with boundary and initial condition losses.

### 3.1.1 RESIDUAL-BASED ATTENTION (RBA)

RBA allocates greater weights to loss terms with larger residuals, emphasizing regions where predictions deviate significantly from true values (Anagnostopoulos et al., 2024). Implemented as a **point-wise tensor**, the RBA weights $w_{i,j}^{\text{RBA}}$ for each loss component $\mathcal{L}_i$ ($i \in \{r, bc, ic\}$) in every point $j$ are updated iteratively:

$$
w_{i,j}^{\text{RBA}} \leftarrow (1-\eta)w_{i,j}^{\text{RBA}} + \eta \left( \frac{|\mathcal{L}_{i,j}|}{\max_j(|\mathcal{L}_{i,j}|)} \right),
\tag{4}
$$

where $\eta$ is the learning rate for RBA weights, and $\max(|\mathcal{L}_{i,j}|)$ is the maximum absolute value of $\mathcal{L}_i$ across the training data $j$.

### 3.1.2 GRADIENT-RELATED ATTENTION (GRA)

GRA dynamically adjusts weights based on gradient norms of different loss components, promoting balanced training. As a **scalar** applied to one entire loss term, GRA addresses the imbalance where gradient norms of the PDE residual loss significantly exceed those of the data fitting loss (Wang et al., 2021), which can lead to pathological gradient flow issues (Wang et al., 2022; Fang et al., 2023). Our mechanism smooths weight adjustments, preventing the network from overemphasizing

residual loss terms and neglecting other essential physical constraints, thus enhancing convergence and stability.

The GRA weight $\lambda^{\text{GRA}}$ is computed as:

$$\hat{\lambda}_{bc,ic}^{\text{GRA}} = \frac{G_r^{\max}}{\epsilon + \overline{G}_{bc,ic}}, \tag{5}$$

where $G_r^{\max} = \max_p \left\| \frac{\partial \mathcal{L}_r}{\partial \theta_p} \right\|$ is the maximum gradient norm of the residual loss, $\overline{G}_i = \frac{1}{P} \sum_{p=1}^{P} \left\| \frac{\partial \mathcal{L}_i}{\partial \theta_p} \right\|$ is the average gradient norm for $\mathcal{L}_i$ ($i \in \{bc, ic\}$), $P$ is the number of model parameters, and $\epsilon$ prevents division by zero.

To smooth the GRA weights over iterations, we apply an exponential moving average:

$$\lambda_{bc,ic}^{\text{GRA}} \leftarrow (1 - \beta_w)\lambda_{bc,ic}^{\text{GRA}} + \beta_w \hat{\lambda}_{bc,ic}^{\text{GRA}}, \tag{6}$$

where $\beta_w$ is the learning rate for the GRA weights. We enforce a minimum value for numerical stability:

$$\lambda_{bc,ic}^{\text{GRA}} \leftarrow \max\left(\lambda_{bc,ic}^{\text{GRA}}, 1 + \epsilon\right). \tag{7}$$

### 3.1.3 COMBINED ATTENTION MECHANISM

To balance the magnitudes of GRA and RBA weights, we apply a logarithmic transformation to the GRA weights when multiplying them with the loss terms, but keep their original form during weight updates. This preserves the direct correlation between weights and gradient information, ensuring sensitivity to discrepancies between residual and data gradients. The logarithmic transformation moderates magnitude differences, preventing imbalances in loss term magnitudes. It allows GRA weights to change more rapidly when discrepancies are small, while ensuring stable updates when discrepancies are large.

By integrating point-wise RBA with term-wise GRA, the total loss under the RGA mechanism is defined as:

$$\mathcal{L}_{\text{RGA}} = \lambda_r w_r^{\text{RBA}} \mathcal{L}_r + \lambda_{bc} w_{bc}^{\text{RBA}} \log\left(\lambda_{bc}^{\text{GRA}}\right) \mathcal{L}_{bc} + \lambda_{ic} w_{ic}^{\text{RBA}} \log\left(\lambda_{ic}^{\text{GRA}}\right) \mathcal{L}_{ic}, \tag{8}$$

where $\lambda_r$, $\lambda_{bc}$, and $\lambda_{ic}$ are scaling factors (typically set to 1), $w^{\text{RBA}}$ are the RBA weights, and $\lambda_i^{\text{GRA}}$ are the GRA weights for $i \in \{bc, ic\}$.

This formulation reweights the residual loss based on its magnitude and adjusts the boundary and initial condition losses according to both their magnitudes and gradient norms, promoting balanced and focused training through a dual attention mechanism. The whole algorithmic details are provided in algorithm 1.

RGA enhances PINNs by dynamically adjusting loss weights based on residual magnitudes and gradient norms. By integrating RBA and GRA, it balances loss contributions, preventing any single component from dominating the training process. This adaptive reweighting accelerates and stabilizes convergence, focusing on challenging regions with significant errors or imbalanced gradients. Consequently, RGA provides a robust framework for more accurate and efficient solutions to complex differential equations, performing well in our AC-PKAN model and potentially benefiting other PINN variants.

## 3.2 CHEBYSHEV1-BASED KOLMOGOROV-ARNOLD NETWORK LAYER

Unlike traditional Kolmogorov-Arnold Networks (KAN) that employ spline coefficients, the *First-kind Chebyshev KAN Layer* leverages the properties of mesh-free Chebyshev polynomials to enhance both computational efficiency and approximation accuracy (SynodicMonth, 2024; Shukla et al., 2024).

Let $\mathbf{x} \in \mathbb{R}^{d_{\text{in}}}$ denote the input vector, where $d_{\text{in}}$ is the input dimensionality, and let $d_{\text{out}}$ be the output dimensionality. Cheby1KAN aims to approximate the mapping $\mathbf{x} \mapsto \mathbf{y} \in \mathbb{R}^{d_{\text{out}}}$ using Chebyshev polynomials up to degree $N$. The Chebyshev polynomials of the first kind, $T_n(x)$, are defined as:

$$T_n(x) = \cos\left(n \arccos(x)\right), \quad x \in [-1, 1], \quad n = 0, 1, \ldots, N. \tag{9}$$

To ensure the input values fall within the domain $[-1, 1]$, Cheby1KAN applies the hyperbolic tangent function for normalization:

$$\tilde{\mathbf{x}} = \tanh(\mathbf{x}). \tag{10}$$

---

**Algorithm 1** Implementation of the RGA Mechanism

---

**Data:** Model parameters $\theta$, total number of parameters $P$, learning rate $\alpha$, hyperparameters $\eta$, $\beta_w$, $\epsilon$
**Initialization:** $w_{r,bc,ic}^{\text{RBA}} \leftarrow 0$, $\lambda_{bc,ic}^{\text{GRA}} \leftarrow 1$

1: **for** each training iteration **do**
2:     Compute gradients:
$$\nabla_\theta \mathcal{L}_i \leftarrow \frac{\partial \mathcal{L}_i}{\partial \theta}, \quad i \in \{r, bc, ic\}$$
3:     Update RBA weights for each data point $j$:
$$w_{i,j}^{\text{RBA}} \leftarrow (1 - \eta) w_{i,j}^{\text{RBA}} + \eta \left( \frac{|\mathcal{L}_{i,j}|}{\max_j |\mathcal{L}_{i,j}|} \right), \quad i \in \{r, bc, ic\}$$
4:     Compute gradient norms:
$$G_r^{\max} \leftarrow \max_p \left\| \nabla_{\theta_p} \mathcal{L}_r \right\|, \quad \overline{G}_i \leftarrow \frac{1}{P} \sum_{p=1}^{P} \left\| \nabla_{\theta_p} \mathcal{L}_i \right\|, \quad i \in \{bc, ic\}$$
5:     Update GRA weights:
$$\hat{\lambda}_i \leftarrow \frac{G_r^{\max}}{\epsilon + \overline{G}_i}, \quad \lambda_i^{\text{GRA}} \leftarrow (1 - \beta_w) \lambda_i^{\text{GRA}} + \beta_w \hat{\lambda}_i, \quad \lambda_i^{\text{GRA}} \leftarrow \max\left(1 + \epsilon, \lambda_i^{\text{GRA}}\right), \quad i \in \{bc, ic\}$$
6:     Compute total loss:
$$\mathcal{L}_{\text{RGA}} \leftarrow \lambda_r w_r^{\text{RBA}} \mathcal{L}_r + \sum_{i \in \{bc, ic\}} \lambda_i w_i^{\text{RBA}} \log\left(\lambda_i^{\text{GRA}}\right) \mathcal{L}_i$$
7:     Update model parameters:
$$\theta \leftarrow \theta - \alpha \nabla_\theta \mathcal{L}_{\text{RGA}}$$
8: **end for**

---

Defining a matrix of functions $\boldsymbol{\Phi}(\tilde{\mathbf{x}}) \in \mathbb{R}^{d_{\text{out}} \times d_{\text{in}}}$, where each element $\Phi_{k,i}(\tilde{x}_i)$ depends solely on the $i$-th normalized input component $\tilde{x}_i$:

$$\Phi_{k,i}(\tilde{x}_i) = \sum_{n=0}^{N} C_{k,i,n} \, T_n(\tilde{x}_i), \quad \text{for } k = 1, 2, \ldots, d_{\text{out}}, \; i = 1, 2, \ldots, d_{\text{in}}. \tag{11}$$

Here, $C_{k,i,n}$ are the learnable coefficients, $T_n(\tilde{x}_i)$ denotes the Chebyshev polynomial of degree $n$ evaluated at $\tilde{x}_i$, and $N$ is the maximum polynomial degree considered.

The output vector $\mathbf{y} \in \mathbb{R}^{d_{\text{out}}}$ is computed by summing over all input dimensions:

$$y_k = \sum_{i=1}^{d_{\text{in}}} \Phi_{k,i}(\tilde{x}_i), \quad \text{for } k = 1, 2, \ldots, d_{\text{out}}, \tag{12}$$

For a network comprising multiple Chebyshev KAN layers, the forward computation can be viewed as a recursive application of this process. Let $\mathbf{x}_l$ denote the input to the $l$-th layer, where $l = 0, 1, \ldots, L - 1$. After applying hyperbolic tangent function to obtain $\tilde{\mathbf{x}}_l = \tanh(\mathbf{x}_l)$, the computation proceeds as follows:

$$\mathbf{x}_{l+1} = \underbrace{\begin{pmatrix} \Phi_{l,1,1}(\cdot) & \Phi_{l,1,2}(\cdot) & \cdots & \Phi_{l,1,n_l}(\cdot) \\ \Phi_{l,2,1}(\cdot) & \Phi_{l,2,2}(\cdot) & \cdots & \Phi_{l,2,n_l}(\cdot) \\ \vdots & \vdots & \ddots & \vdots \\ \Phi_{l,n_{l+1},1}(\cdot) & \Phi_{l,n_{l+1},2}(\cdot) & \cdots & \Phi_{l,n_{l+1},n_l}(\cdot) \end{pmatrix}}_{\boldsymbol{\Phi}_l} \tilde{\mathbf{x}}_l, \tag{13}$$

A general cheby1KAN network is a composition of $L$ layers: given an input vector $\mathbf{x}_0 \in \mathbb{R}^{n_0}$, the overall output of the KAN network is:

$$\text{Cheby1KAN}(\mathbf{x}) = (\boldsymbol{\Phi}_{L-1} \circ \boldsymbol{\Phi}_{L-2} \circ \cdots \circ \boldsymbol{\Phi}_1 \circ \boldsymbol{\Phi}_0)\mathbf{x}. \tag{14}$$

According to the original author's recommended configuration (SynodicMonth, 2024), we also apply LayerNorm after Cheby1KAN Layer to prevent gradient vanishing induced by the use of tanh.

### 3.3 INTERNAL MODEL ARCHITECTURE

We present the internal architecture of the proposed *AC-PKAN* model, which enhances the framework with an advanced attention mechanism (Wang et al., 2021; 2024a). To effectively capture complex relationships within the data, the internal attention enhanced architecture integrates linear input upscaling and output downscaling layers, adaptive activation functions, and Cheby1KAN layers. The detailed architecture is outlined in Algorithm 2.

**Linear Upscaling and Downscaling Layers**   To adjust data dimensionality, the model employs linear transformations at the input and output stages. The input features $\mathbf{x}$ are projected into a higher-dimensional space, and the final network representation $\alpha^{(L)}$ is mapped to the output space via:

$$\mathbf{h}_0 = \mathbf{W}_{\text{emb}}\mathbf{x} + \mathbf{b}_{\text{emb}}, \quad \mathbf{y} = \mathbf{W}_{\text{out}}\alpha^{(L)} + \mathbf{b}_{\text{out}}, \tag{15}$$

where $\mathbf{W}_{\text{emb}} \in \mathbb{R}^{d_{\text{model}} \times d_{\text{in}}}$, $\mathbf{b}_{\text{emb}} \in \mathbb{R}^{d_{\text{model}}}$, $\mathbf{W}_{\text{out}} \in \mathbb{R}^{d_{\text{out}} \times d_{\text{hidden}}}$, and $\mathbf{b}_{\text{out}} \in \mathbb{R}^{d_{\text{out}}}$ are learnable parameters.

**Adaptive Activation Function**   We employ a novel activation function called *Wavelet*, inspired by Fourier transforms, to introduce non-linearity and capture periodic patterns effectively (Zhao et al., 2023):

$$\text{Wavelet}(x) = w_1 \sin(x) + w_2 \cos(x), \tag{16}$$

where $w_1$ and $w_2$ are learnable parameters initialized to one.

**Attention Mechanism**   An internal attention mechanism is incorporated by computing two feature representations, $\mathbf{U}$ and $\mathbf{V}$, via the *Wavelet* activation applied to linear transformations of the embedded inputs:

$$\mathbf{U} = \text{Wavelet}(\mathbf{h}_0\boldsymbol{\Theta}_U + \mathbf{b}_U), \quad \mathbf{V} = \text{Wavelet}(\mathbf{h}_0\boldsymbol{\Theta}_V + \mathbf{b}_V), \tag{17}$$

where $\boldsymbol{\Theta}_U, \boldsymbol{\Theta}_V \in \mathbb{R}^{d_{\text{model}} \times d_{\text{hidden}}}$ and $\mathbf{b}_U, \mathbf{b}_V \in \mathbb{R}^{d_{\text{hidden}}}$ are learnable parameters.

**Attention Integration**   The attention mechanism integrates $\mathbf{U}$ and $\mathbf{V}$ iteratively across layers using the following equations:

$$\alpha_0^{(l)} = \mathbf{H}^{(l)} + \alpha^{(l-1)}, \quad \alpha^{(l)} = (1 - \alpha_0^{(l)}) \odot \mathbf{U} + \alpha_0^{(l)} \odot (\mathbf{V} + 1), \tag{18}$$

where $\alpha^{(0)} = \mathbf{U}$ and $\odot$ denotes element-wise multiplication. Here, $\mathbf{H}^{(l)} \in \mathbb{R}^{N \times d_{\text{hidden}}}$ is the output of the $l$-th Cheby1KAN layer after LayerNormalization, and $N$ is the number of nodes.

---

**Algorithm 2** Internal AC-PKAN Forward Pass

---

**Data:** Input data $\mathbf{x}$, Cheby1KAN layer parameters, Wavelet activation function parameters
**Initialization:** Randomly initialize weights $\mathbf{W}_{\text{emb}}, \boldsymbol{\Theta}_U, \boldsymbol{\Theta}_V, \mathbf{W}_{\text{out}}$ and biases $\mathbf{b}_{\text{emb}}, \mathbf{b}_U, \mathbf{b}_V, \mathbf{b}_{\text{out}}$

1: Input embedding:
$$\mathbf{h}_0 \leftarrow \mathbf{W}_{\text{emb}}\mathbf{x} + \mathbf{b}_{\text{emb}}$$

2: Compute representations:
$$\mathbf{U} \leftarrow \text{Wavelet}(\mathbf{h}_0\boldsymbol{\Theta}_U + \mathbf{b}_U), \quad \mathbf{V} \leftarrow \text{Wavelet}(\mathbf{h}_0\boldsymbol{\Theta}_V + \mathbf{b}_V)$$

3: Initialize attention:
$$\alpha^{(0)} \leftarrow \mathbf{U}$$

4: **for** $l = 1$ to $L$ **do**
5:     Update attention:
$$\mathbf{H}^{(l)} \leftarrow \text{LayerNorm}\left(\text{Cheby1KANLayer}\left(\alpha^{(l-1)}\right)\right)$$

$$\alpha_0^{(l)} \leftarrow \mathbf{H}^{(l)} + \alpha^{(l-1)}$$
$$\alpha^{(l)} \leftarrow (1 - \alpha_0^{(l)}) \odot \mathbf{U} + \alpha_0^{(l)} \odot (\mathbf{V} + 1)$$

6: **end for**
7: Output prediction:
$$\mathbf{y} \leftarrow \mathbf{W}_{\text{out}}\alpha^{(L)} + \mathbf{b}_{\text{out}}$$

---

**Approximation Ability** Our AC-PKAN's inherent attention mechanism eliminates the need for an additional bias function $b(x)$ required in previous KAN models to maintain non-zero higher-order derivatives (Wang et al., 2024b). This reduces model complexity and parameter count while preserving the ability to seamlessly approximate PDEs of arbitrary finite order. By ensuring non-zero derivatives of any finite order and invoking the Kolmogorov–Arnold representation theorem, our model can approximate such PDEs.

**Proposition 1.** *Let $\mathcal{N}$ be an AC-PKAN model with $L$ layers ($L \geq 2$) and infinite width. Then, the output $y = \mathcal{N}(x)$ has non-zero derivatives of any finite-order with respect to the input $x$.*

*Proof sketch:* The attention mechanism and sinusoidal activations in AC-PKAN ensure that the output function has non-zero derivatives of all orders, enabling the approximation of high-order PDEs without additional bias functions; the full proof is provided in Appendix A.

By combining our AC-PKAN internal architecture with the external RGA mechanism, we obtain the complete AC-PKAN model. Figure 1 provides a detailed illustration of our model structure.

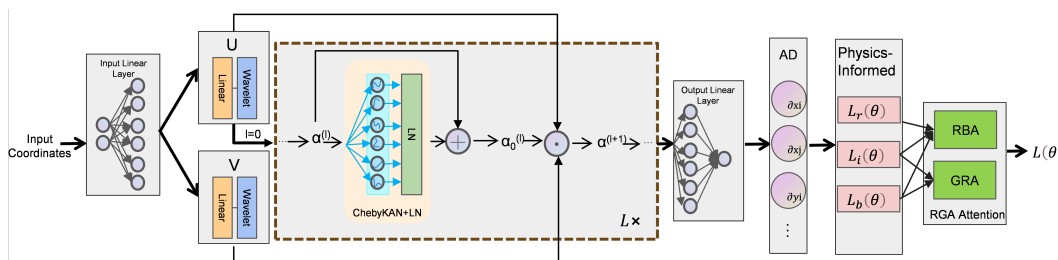

Figure 1: Architecture of the complete AC-PKAN model. It combines its internal attention architecture with an external attention strategy, yielding a weighted loss optimized to obtain the predicted solution $u$.

## 4 EXPERIMENTS

**Goal.** Our empirical evaluations aim to demonstrate three key advantages of the AC-PKAN model: (1) its intrinsic structure exhibits powerful symbolic representation and function approximation capabilities even without the RGA loss weighting mechanism; (2) it significantly improves generalization abilities and mitigates failure modes compared to PINNs and other KAN variants; and (3) it achieves superior performance in complex real-world engineering environments. To validate these claims, we designed three categories of experimental tasks across five experiments, comparing 12 models including PINN, PINNsFormer, KAN, and fKAN baselines. The experimental setup was inspired by methodologies in (SynodicMonth, 2024; Hao et al., 2023; Wang et al., 2024b; Zhao et al., 2023; Wang et al., 2023). The code for our model and experiments will be made publicly available upon acceptance of this paper.

### 4.1 COMPLEX FUNCTION FITTING

We evaluated our AC-PKAN Simplified model—which employs only the internal architecture—against PINN (MLP), KAN, and various KAN variants on a complex function interpolation task. Detailed experimental setups and results are provided in Appendices D and E.

As shown in Figure 2, the AC-PKAN Simplified model converges more rapidly than MLPs, KAN, and most KAN variants, achieving lower final losses. While Cheby2KAN and FourierKAN demonstrate faster convergence, our model produces smoother fitted curves and exhibits greater robustness to noise, effectively preventing overfitting in regions with high-frequency variations. Performance metrics are presented in Table 1.

| Model | rMAE | rMSE | Loss |
|---|---|---|---|
| Cheby1KAN | 0.0179 | 0.0329 | **0.0068** |
| Cheby2KAN | 0.0189 | 0.0313 | 0.0079 |
| MLP | 0.0627 | 0.1250 | 0.1410 |
| AC-PKAN_s | **0.0177** | **0.0311** | 0.0081 |
| KAN | *0.0145* | *0.0278* | 0.0114 |
| rKAN | 0.0458 | 0.0783 | 0.1867 |
| fKAN | 0.0858 | 0.1427 | 0.1722 |
| FastKAN | 0.0730 | 0.1341 | 0.1399 |
| FourierKAN | 0.0211 | 0.0353 | *0.0063* |

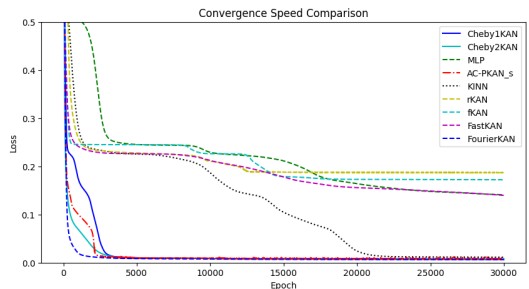

Table 1: Comparison of test rMAE, rMSE, and training Loss among Models

Figure 2: Convergence Comparison of Nine Different Models

### 4.2 MITIGATING FAILURE MODES IN PINNS

We assessed the AC-PKAN model on two complex PDEs known as PINN failure modes—the 1D-reaction, 1D-wave and 2D Navier–Stokes equations (Mojgani et al., 2022; Daw et al., 2022; Krishnapriyan et al., 2021)—to demonstrate its superior generalization ability compared to other PINN variants. In these cases, optimization often becomes trapped in local minima, leading to overly smooth approximations that deviate from true solutions.

Evaluation results are summarized in Table 2, with detailed PDE formulations and setups in Appendix D. Prediction and absolute error plots for AC-PKAN and KINN are shown in Figure 3a and 3b; additional plots are in Appendix E.

AC-PKAN significantly outperforms nearly all baselines, achieving the lowest or second-lowest test errors, thus more effectively mitigating failure modes than the previous state-of-the-art method, PINNsFormer. Other baselines remain stuck in local minima, failing to optimize the loss effectively. These results highlight the advantages of AC-PKAN in generalization and approximation accuracy over conventional PINNs, KANs, and existing variants.

We also plotted the mean values of the RBA and GRA weights over epochs in Figure 3c. The mean RBA weights for all loss terms eventually converge, indicating mitigation of residual imbalance. In contrast, the GRA weights continue to increase, suggesting persistent gradient imbalance. The

| Model | 2D-NS | | 1D-Wave | | 1D-Reaction | |
|---|---|---|---|---|---|---|
| | rMAE | rRMSE | rMAE | rRMSE | rMAE | rRMSE |
| PINN | 3.6949 | 3.2899 | 0.3182 | 0.3200 | 0.9818 | 0.9810 |
| QRes | 3.2930 | 3.6998 | 0.3507 | 0.3485 | 0.9844 | 0.9849 |
| FLS | 3.6930 | 3.2893 | 0.3810 | 0.3796 | 0.9793 | 0.9773 |
| PINNsFormer | 3.6986 | 3.2924 | **0.2699** | **0.2825** | *0.0152* | *0.0300* |
| Cheby1KAN | 3.7561 | 3.3347 | 1.1240 | 1.0866 | 0.0617 | 0.1329 |
| Cheby2KAN | **3.0443** | 2.9513 | 1.1239 | 1.0865 | 1.0387 | 1.0256 |
| AC-PKAN | *2.4519* | 2.4412 | **0.0011** | **0.0011** | **0.0375** | **0.0969** |
| KINN | 3.6816 | 3.2801 | 0.3466 | 0.3456 | 0.1314 | 0.2101 |
| rKAN | NaN | NaN | 247.7560 | 2593.0750 | 65.2014 | 54.8567 |
| FastKAN | 3.6999 | *1.3401* | 0.5312 | 0.5229 | 0.5475 | 0.6030 |
| fKAN | 3.7040 | 3.2998 | 0.4884 | 0.4768 | 0.0604 | 0.1033 |
| FourierKAN | 5672.3763 | 5973.1545 | 1.1356 | 1.1018 | 1.4542 | 1.4217 |

Table 2: Experimental results demonstrating that our AC-PKAN model achieves best or second-best performance on three challenging PDE tasks.

| Model | Heterogeneous Problem | | Complex Geometry | |
|---|---|---|---|---|
| | rMAE | rRMSE | rMAE | rRMSE |
| PINNs | 0.1662 | 0.1747 | 0.9010 | 0.9289 |
| QRes | 0.1102 | *0.1140* | 0.9024 | 0.9289 |
| FLS | 0.1701 | 0.1789 | 0.9021 | 0.9287 |
| PINNsFormer | *0.1008* | **0.1610** | **0.8851** | **0.8721** |
| Cheby1KAN | 0.1404 | 0.2083 | 0.9026 | 0.9244 |
| Cheby2KAN | 0.4590 | 0.5155 | 0.9170 | 1.0131 |
| AC-PKAN | **0.1063** | 0.1817 | *0.5452* | *0.5896* |
| KINN | 0.1599 | 0.1690 | 0.9029 | 0.9261 |
| rKAN | 24.8319 | 380.5582 | 23.5426 | 215.4764 |
| FastKAN | 0.1549 | 0.1624 | 0.9034 | 0.9238 |
| fKAN | 0.1179 | 0.1724 | 0.9043 | 0.9303 |
| FourierKAN | 0.4588 | 0.5154 | 1.4455 | 1.5341 |

Table 3: Experimental results demonstrating that AC-PKAN achieves the best or second-best performance across two complex environmental PDE tasks.

steadily increasing GRA weights effectively alleviate the gradient stiffness problem, consistent with findings in (Wang et al., 2021). The significant magnitude discrepancy between GRA and RBA data supports using a logarithmic function for GRA weights in loss weighting.

Additionally, integrating AC-PKAN with other external learning strategies, such as the Neural Tangent Kernel (NTK) method, resulted in enhanced performance (Table 4). This demonstrates the flexibility of AC-PKAN in incorporating various learning schemes, offering practical and customizable solutions for accurate modeling in real-world applications.

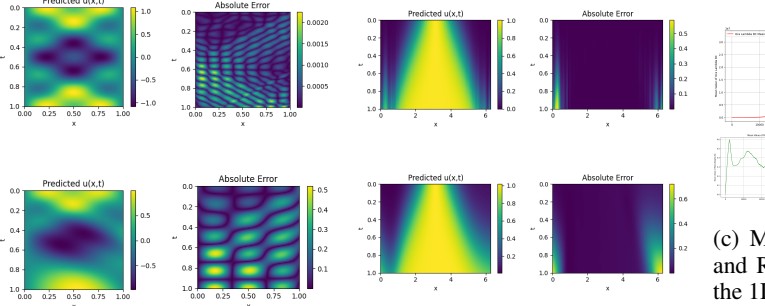

(a) Predictions and errors for the 1D-Wave equation.

(b) Predictions and errors for the 1D-Reaction equation.

(c) Mean values of GRA weights and RBA weights over epochs for the 1D-Wave experiment.

Figure 3: Subfigures (a) and (b) show predictions and absolute errors for the 1D-Wave and 1D-Reaction equations, with AC-PKAN results above and KINN below. Subfigure (c) presents the mean GRA and RBA weights during the 1D-Wave experiment.

| Model | rMAE | rRMSE |
|---|---|---|
| AC-PKAN + *NTK* | **0.0009** | **0.0009** |
| PINNs + *NTK* | 0.1397 | 0.1489 |
| PINNsFormer + *NTK* | 0.0453 | 0.0484 |

Table 4: Performance comparison on the 1D-wave equation using the NTK method. AC-PKAN combined with NTK achieves superior results across all metrics.

| Model | rMAE | rRMSE |
|---|---|---|
| **AC-PKAN** | **0.0011** | **0.0011** |
| AC-PKAN (no GRA) | 0.0779 | 0.0787 |
| AC-PKAN (no RBA) | 0.0494 | 0.0500 |
| AC-PKAN (no RGA) | 0.4549 | 0.4488 |
| AC-PKAN (no Wavelet) | 0.0045 | 0.0046 |
| AC-PKAN (no Encoder) | 0.0599 | 0.0584 |
| AC-PKAN (no Linear) | 1.0422 | 1.0246 |

Table 5: Ablation study for 1D-Wave demonstrating the impact of each module on the performance of AC-PKAN.

## 4.3 PDEs in Complex Engineering Environments

We further tested AC-PKAN in two challenging scenarios: heterogeneous environments and complex geometric boundary conditions. Literature indicates that PINNs struggle with heterogeneous

problems due to sensitivity to material properties (Aliakbari et al., 2023), significant errors near boundary layers (Piao et al., 2024), and convergence issues (Sumanta et al., 2024). Original KANs also perform poorly with complex geometries (Wang et al., 2024b). We applied AC-PKAN to solve Poisson equations in these environments.

Detailed PDE formulations are in Appendix D, and detailed experimental results are illustrated in Appendix E. Summarized in Table 3 and partially shown in Figure 4, the results indicate that AC-PKAN consistently achieves the best or second-best performance. It demonstrates superior potential in solving heterogeneous problems without subdomain division and exhibits promising application potential in complex geometric boundary problems where most models fail.

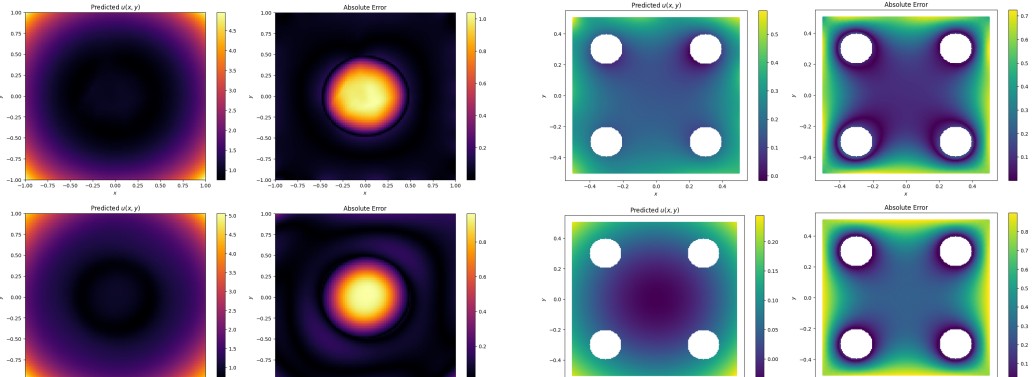

Figure 4: Predictions and absolute errors for the Heterogeneous Problem (left) and Complex Geometry (right). In each panel, the top two images show AC-PKAN results, and the bottom two images show PINNsformer results.

### 4.4 ADDITIONAL EXPERIMENTS

**Loss Landscape Analysis**    Figure 5 shows that the loss landscapes of PINNsFormer, fKAN, and QRes are more complex than that of AC-PKAN. Although Cheby1KAN appears to have a simpler loss landscape, its steep gradients hinder optimization. Except for AC-PKAN, other models display multiple local minima near the optimal point, increasing convergence difficulty.

**Ablation Study**    Ablation experiments on the 1D-Wave equation (Table 5) confirm that each module in our model is crucial. Removing any module leads to a significant performance decline, especially the Linear module. These findings suggest that the KAN architecture alone is insufficient for complex tasks, validating our integration of MLPs with the Cheby1KAN framework.For additional ablation studies, please refer to Section C.

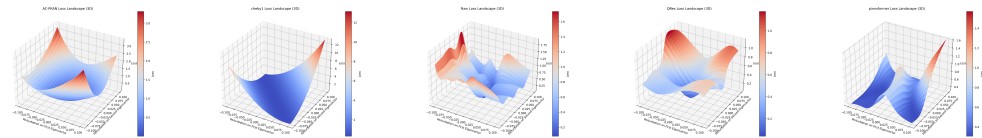

Figure 5: Loss landscapes on the 1D-Wave experiment of various models (from left to right): AC-PKAN, Cheby1KAN, fkan, QRes, and Pinnsformer.

## 5 CONCLUSION

We introduced AC-PKAN, a novel framework that enhances PINNs by integrating Cheby1KAN with traditional MLPs and augmenting them with internal and external attention mechanisms. This improves the model's ability to capture complex patterns and dependencies, resulting in superior performance on challenging PDE tasks, including previous PINN failure modes and complex physical environments.

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

## A   PROOF OF THEOREM 1

**Theorem 1.** *Let $\mathcal{N}$ be an AC-PKAN model with $L$ layers ($L \geq 2$) and infinite width. Then, the output $y = \mathcal{N}(x)$ has non-zero derivatives of any finite-order with respect to the input $x$.*

*Proof.* Consider the forward propagation process of the AC-PKAN. We begin with the initial layer:

$$h_0 = W_{\text{emb}}x + b_{\text{emb}}, \tag{19}$$

$$U = \omega_{U,1} \sin(h_0\theta_U + b_U) + \omega_{U,2} \cos(h_0\theta_U + b_U), \tag{20}$$

$$V = \omega_{V,1} \sin(h_0\theta_V + b_V) + \omega_{V,2} \cos(h_0\theta_V + b_V), \tag{21}$$

$$\alpha^{(0)} = U. \tag{22}$$

For each layer $l = 1, 2, \ldots, L$, the computations proceed as follows:

$$H^{(l)} = \sum_{i=1}^{d_{\text{in}}} \sum_{k=1}^{d_{\text{out}}} \sum_{n=0}^{N} C_{k,i,n} T_n \left( \tanh(\alpha^{(l-1)}) \right), \tag{23}$$

$$\alpha_0^{(l)} = H^{(l)} + \alpha^{(l-1)}, \tag{24}$$

$$\alpha^{(l)} = (1 - \alpha_0^{(l)}) \odot U + \alpha_0^{(l)} \odot (V + 1), \tag{25}$$

$$y = W_{\text{out}}\alpha^{(L)} + b_{\text{out}}. \tag{26}$$

During the backward propagation, we derive the derivative of the output with respect to the input $x$, which approximates the differential operator of the PDEs. Focusing on the first-order derivative as an example:

$$\frac{\partial y}{\partial x} = \frac{\partial y}{\partial \alpha^{(L)}} \frac{\partial \alpha^{(L)}}{\partial x}$$

$$= W_{\text{out}} \frac{\partial \alpha^{(L)}}{\partial x}. \tag{27}$$

Expanding $\frac{\partial \alpha^{(L)}}{\partial x}$:

$$\frac{\partial \alpha^{(L)}}{\partial x} = -\frac{\partial \alpha_0^{(L)}}{\partial x} \odot U + \left(1 - \alpha_0^{(L)}\right) \odot \frac{\partial U}{\partial x} + \frac{\partial \alpha_0^{(L)}}{\partial x} \odot (V + 1) + \alpha_0^{(L)} \odot \frac{\partial V}{\partial x}$$

$$= \frac{\partial \alpha_0^{(L)}}{\partial x} \odot (V - U + 1) + \alpha_0^{(L)} \odot \left( \frac{\partial V}{\partial x} - \frac{\partial U}{\partial x} \right) + \frac{\partial U}{\partial x}$$

$$= \left( \frac{\partial H^{(L)}}{\partial x} + \frac{\partial \alpha^{(L-1)}}{\partial x} \right) \odot (V - U + 1) + \left( H^{(L)} + \alpha^{(L-1)} \right) \odot \left( \frac{\partial V}{\partial x} - \frac{\partial U}{\partial x} \right) + \frac{\partial U}{\partial x}. \tag{28}$$

This establishes a recursive relationship for the derivatives. Define:

$$A^{(l)} = \frac{\partial H^{(l)}}{\partial x} + \frac{\partial \alpha^{(l-1)}}{\partial x}, \tag{29}$$

$$B^{(l)} = H^{(l)} + \alpha^{(l-1)}. \tag{30}$$

for each layer $l = 1, 2, \ldots, L$.

For the base case $l = 1$:

$$A^{(1)} = \frac{\partial H^{(1)}}{\partial x} + \frac{\partial \alpha^{(0)}}{\partial x}$$

$$= \left( \sum_{i=1}^{d_{\text{in}}} \sum_{k=1}^{d_{\text{out}}} \sum_{n=0}^{N} C_{k,i,n} T'_n \left( \tanh(\alpha^{(0)}) \right) \text{sech}^2(\alpha^{(0)}) + 1 \right) \frac{\partial \alpha^{(0)}}{\partial x}, \tag{31}$$

$$\frac{\partial \alpha^{(0)}}{\partial x} = \frac{\partial U}{\partial x}$$

$$= W_{\text{emb}}\theta_U \left[ \omega_{U,1} \cos(h_0\theta_U + b_U) - \omega_{U,2} \sin(h_0\theta_U + b_U) \right] \neq 0, \tag{32}$$

Moreover,

$$B^{(1)} = H^{(1)} + \alpha^{(0)}$$

$$= \sum_{i=1}^{d_{\text{in}}} \sum_{k=1}^{d_{\text{out}}} \sum_{n=0}^{N} C_{k,i,n} T_n \left( \tanh(\alpha^{(0)}) \right) + \alpha^{(0)}. \tag{33}$$

For layers $l > 1$, where $l \in \mathbb{N}^*$:

$$A^{(l)} = \left( \sum_{i=1}^{d_{\text{in}}} \sum_{k=1}^{d_{\text{out}}} \sum_{n=0}^{N} C_{k,i,n} T_n' \left( \tanh(\alpha^{(l-1)}) \right) \operatorname{sech}^2(\alpha^{(l-1)}) + 1 \right) \frac{\partial \alpha^{(l-1)}}{\partial x}. \tag{34}$$

We have established a recursive relationship.

Notably, the first derivative of the Chebyshev polynomial is given by

$$T_n'(x) = \frac{d}{dx} T_n(x) = \frac{n \sin(n \arccos(x))}{\sqrt{1 - x^2}},$$

and higher-order derivatives satisfy

$$T_n^{(k)}(x) = 0 \quad \text{for all } k > n.$$

Therefore, for any order $k > n$, the $k$-th derivative of $A^{(l)}$ is identically zero. Consequently, the $k$-th derivative of the first part of equation 28 is zero.

However, observe that:

$$B^{(l)} = \sum_{i=1}^{d_{\text{in}}} \sum_{k=1}^{d_{\text{out}}} \sum_{n=0}^{N} C_{k,i,n} T_n \left( \tanh(\alpha^{(l-1)}) \right) + \alpha^{(l-1)}, \tag{35}$$

since the derivatives of $\alpha^{(l-1)}$ for any finite order are non-zero, the derivatives of $B^{(l)}$ are non-zero.

Furthermore, we have:

$$\frac{\partial V}{\partial x} - \frac{\partial U}{\partial x} = W_{\text{emb}} \left( \theta_V \left[ \omega_{V,1} \cos(h_0 \theta_V + b_V) - \omega_{V,2} \sin(h_0 \theta_V + b_V) \right] \right.$$
$$\left. - \theta_U \left[ \omega_{U,1} \cos(h_0 \theta_U + b_U) - \omega_{U,2} \sin(h_0 \theta_U + b_U) \right] \right), \tag{36}$$

the derivatives $\frac{\partial V}{\partial x} - \frac{\partial U}{\partial x}$ of any finite order are also non-zero. Additionally, the third component of equation 28, $\frac{\partial U}{\partial x}$, is non-zero. Therefore, the $k$-th derivatives of the remaining parts of equation 28 are non-zero. Thus the $k$-th derivatives of equation 27 are non-zero.

Consequently, for any positive integer $N$, the $N$-th derivative of the output with respect to the input $\frac{\partial^N y}{\partial x^N}$ exists and is non-zero. This guarantees that the AC-PKAN possesses the capacity to approximate PDEs of arbitrary high order. $\qquad\square$

# B  EXPLANATION FOR THE SUPERIORITY OF CHEBYSHEV TYPE I POLYNOMIALS OVER B-SPLINES

Chebyshev polynomials of the first kind, defined by $T_n(x) = \cos(n \arccos(x))$, concentrate their spectrum on high frequencies, with the frequency increasing linearly with the polynomial order $n$. This property makes them particularly suitable for capturing high-frequency oscillations, as their high-frequency components decay slowly. The even distribution of extrema further aids in capturing rapid variations, which is beneficial for representing high-frequency features. In contrast, B-splines, being piecewise polynomials, exhibit a rapidly decaying spectrum, limiting their ability to capture high-frequency features effectively.

Chebyshev polynomials possess both global support and global orthogonality over the interval $[-1, 1]$. The value of a Chebyshev polynomial at any point depends on all points within the interval, making them highly effective at capturing global features and high-frequency components. They satisfy the orthogonality relation:

$$\int_{-1}^{1} \frac{T_m(x) T_n(x)}{\sqrt{1 - x^2}} \, dx = 0, \quad \text{for } m \neq n.$$

This orthogonality allows Chebyshev polynomials to achieve minimax approximation, minimizing the maximum error over the interval. In contrast, B-splines have local support; each basis function is nonzero only within a specific subinterval. This local nature limits their ability to capture global high-frequency features. Additionally, B-splines lack global orthogonality, reducing their efficiency in approximating functions.

B-spline-based Kernel Adaptive Networks (KANs) require substantial memory due to the storage of grids and coefficient matrices that scale cubically with grid size and spline order. They store grids of size (in_features, grid_size + 2 × spline_order + 1) and coefficient matrices of size (out_features, in_features, grid_size + spline_order). Since B-splines are piecewise polynomials, each segment requires maintaining basis function values and performing high-order interpolation within its support interval. This involves generating polynomial bases, solving linear systems (e.g., using `torch.linalg.lstsq`), and executing recursive updates, resulting in high computational and storage demands.

In contrast, Chebyshev polynomials are globally defined and require only a coefficient matrix of size (input_dim, output_dim, degree + 1), eliminating terms directly related to grid size and spline order. The memory complexity grows linearly with the degree. Chebyshev polynomials eliminate the need for grid storage and do not require solving linear systems, interpolation, or recursive updates of piecewise basis functions, which significantly reduce computational and storage requirements.

## C  ADDITIONAL ABLATION STUDIES

### C.1  EFFECT OF LOGARITHMIC TRANSFORMATION IN THE RGA MODULE

In this ablation study, we investigated the impact of removing the logarithmic transformation in the RGA module across five PDE experimental tasks. To compensate for the absence of the logarithmic scaling, we adjusted the scaling factors to smaller values. Specifically, we employed the original RGA design to pre-train the models for several epochs, during which very large values of $\lambda^{\mathrm{GRA}}$ were obtained. To maintain consistency in the magnitudes of different loss terms, we set the scaling factor of the PDE residual loss term to 1 and assigned the scaling factors of the data loss terms—including boundary conditions (BC) and initial conditions (IC)—to the negative order of magnitude of the current $\lambda^{\mathrm{GRA}}$.

The performance metrics without the logarithmic transformation are summarized in Table 6.

| Equation | rMAE | rRMSE |
|---|---|---|
| 2D Navier–Stokes | 84.3943 | 88.7684 |
| 1D Wave | 0.7686 | 0.7479 |
| 1D Reaction | 2.2348 | 2.2410 |
| Heterogeneous Problem | 10.0849 | 9.6492 |
| Complex Geometry | 164.4283 | 158.7840 |

Table 6: Performance metrics after removing the logarithmic transformation in the RGA module.

Comparing these results with those in Tables 2 and 3, we observe a significant deterioration in the performance of AC-PKAN when the logarithmic transformation is removed. This decline is attributed to two main factors: first, $\lambda^{\mathrm{GRA}}$ attains excessively large values; second, it exhibits a wide range of variation. During the standard training process, the coefficient $\lambda^{\mathrm{GRA}}$ rapidly grows from 0 to a very large value, resulting in a broad dynamic range. The logarithmic transformation effectively narrows this range; for instance, in the 1D Wave experiment, the scale of $\lambda^{\mathrm{GRA}}$ over epochs ranges from 0 to $4 \times 10^7$, whereas $\ln(\lambda^{\mathrm{GRA}})$ ranges from 7 to 15 in Picture 6. Removing the logarithmic transformation and attempting to manually adjust scaling factors to match the apparent magnitudes is ineffective. The model cannot adapt to the drastic changes in $\lambda^{\mathrm{GRA}}$, and rigid manual scaling factors exacerbate the imbalance among loss terms, ultimately leading to training failure. By confining the variation range of $\lambda^{\mathrm{GRA}}$, the logarithmic transformation enables the model to adjust more flexibly and effectively.

The rationale for employing the logarithmic transformation stems from the Bode plot in control engineering, which uses a logarithmic frequency axis while directly labeling actual frequency values.

This approach not only compresses a wide frequency range but also linearizes the system's gain and phase characteristics on a logarithmic scale, thereby mitigating imbalances caused by significant differences in data scales.

## C.2 Effect of the RGA Module in Other PINN Variants

In this ablation study, we applied our RGA module to other algorithms to assess its generalizability. The experimental results are presented in Table 7.

| Model | rMAE | rRMSE |
|-------|------|-------|
| PINN+RGA | 0.0914 | 0.0924 |
| PINNsFormer+RGA | NaN | NaN |
| QRes+RGA | 0.2204 | 0.2184 |
| FLS+RGA | 0.1610 | 0.1617 |
| Cheby1KAN+RGA | 0.0567 | 0.0586 |
| Cheby2KAN+RGA | 1.0114 | 1.0048 |
| AC-PKAN | **0.0011** | **0.0011** |
| KINN+RGA | *0.0479* | *0.0486* |
| rKAN+RGA | NaN | NaN |
| FastKAN+RGA | 0.1348 | 0.1376 |
| fKAN+RGA | 0.2177 | 0.2149 |
| FourierKAN+RGA | 1.0015 | 1.0001 |

Table 7: Performance metrics of various models with the RGA module applied.

In the case of PINNsFormer+RGA, the results are NaN due to a CUDA out-of-memory error during training. This occurs because PINNsFormer needs to create pseudo sequences, and applying the RGA module—which requires gradient computation on a large number of data points within the pseudo sequence—incurs significant memory overhead, leading to training failure. Meanwhile, rKAN+RGA resulted in NaN due to gradient instability during training.

Excluding these cases, and compared to the results in Tables 2 and 3, the performance of other models improved significantly when incorporating the RGA module. This indicates that our RGA can be generally transferred to other models to enhance their performance. However, it is noteworthy that none of the other models surpassed the performance of our AC-PKAN.

## D Experiment Setup Details

We utilize the AdamW optimizer with a learning rate of $1 \times 10^{-4}$ and a weight decay of $1 \times 10^{-4}$ in all experiments. Meanwhile, all experiments were conducted on an NVIDIA A100 GPU with 40GB of memory. And Xavier initialization is applied to all layers. In PDE-Solving problems, We present the detailed formula of rMAE and rRMSE as the following:

$$\text{rMAE} = \frac{\sum_{n=1}^{N} |\hat{u}(x_n, t_n) - u(x_n, t_n)|}{\sum_{n=1}^{N_{res}} |u(x_n, t_n)|}$$

$$\text{rRMSE} = \sqrt{\frac{\sum_{n=1}^{N} |\hat{u}(x_n, t_n) - u(x_n, t_n)|^2}{\sum_{n=1}^{N} |u(x_n, t_n)|^2}} \tag{37}$$

where $N$ is the number of testing points, $\hat{u}$ is the neural network approximation, and $u$ is the ground truth. The specific details for each experiment are provided below. For further details, please refer to our experiment code repository to be released.

## D.1 Complex Function Fitting Experiment Setup Details

The aim of this experiment is to evaluate the interpolation capabilities of several neural network architectures, including AC-PKAN, Chebyshev-based KAN (ChebyKAN), traditional MLP, and other

advanced models. The task involves approximating a target noisy piecewise 1D function, defined over three distinct intervals.

**Target Function**   The target function $f(x)$ is defined piecewise as follows:

$$
f(x) = \begin{cases}
\sin(25\pi x) + x^2 + 0.5\cos(30\pi x) + 0.2x^3 & x < 0.5, \\
0.5xe^{-x} + |\sin(5\pi x)| + 0.3x\cos(7\pi x) + 0.1e^{-x^2} & 0.5 \le x < 1.5, \\
\frac{\ln(x-1)}{\ln(2)} - \cos(2\pi x) + 0.2\sin(8\pi x) + \frac{0.1\ln(x+1)}{\ln(3)} & x \ge 1.5,
\end{cases}
$$

with added Gaussian noise $\epsilon \sim \mathcal{N}(0, 0.1)$.

**Dataset**

- **Training Data**: 500 points uniformly sampled from the interval $x \in [0, 2]$, with corresponding noisy function values $y = f(x) + \epsilon$.

- **Testing Data**: 1000 points uniformly sampled from the same interval $x \in [0, 2]$ to assess the models' interpolation performance.

**Training Details**

- **Epochs**: Each model is trained for 30,000 epochs.

- **Loss Function**: The Mean Squared Error (MSE) loss is utilized to compute the discrepancy between predicted and true function values:

$$
\mathcal{L}_{\text{MSE}} = \frac{1}{N} \sum_{i=1}^{N} (y_i - \hat{y}_i)^2
$$

- **Weight Initialization**: Xavier initialization is applied to all linear layers.

**Model Hyperparameters**   The parameter counts for each model are summarized in Table 8.

### D.2   FAILURE MODES IN PINNs EXPERIMENT SETUP DETAILS

We selected the one-dimensional wave equation (1D-Wave) and the one-dimensional reaction equation (1D-Reaction) as representative experimental tasks to investigate failure modes in Physics-Informed Neural Networks (PINNs). Below, we provide a comprehensive description of the experimental details, including the formulation of partial differential equations (PDEs), data generation processes, model architecture, training regimen, and hyperparameter selection.

**1D-Wave PDE.**   The 1D-Wave equation is a hyperbolic PDE that is used to describe the propagation of waves in one spatial dimension. It is often used in physics and engineering to model various wave phenomena, such as sound waves, seismic waves, and electromagnetic waves. The system has the formulation with periodic boundary conditions as follows:

$$
\frac{\partial^2 u}{\partial t^2} - \beta \frac{\partial^2 u}{\partial x^2} = 0 \quad \forall x \in [0,1],\ t \in [0,1]
$$

$$
\texttt{IC}: u(x,0) = \sin(\pi x) + \frac{1}{2}\sin(\beta\pi x), \quad \frac{\partial u(x,0)}{\partial t} = 0 \tag{38}
$$

$$
\texttt{BC}: u(0,t) = u(1,t) = 0
$$

where $\beta$ is the wave speed. Here, we are specifying $\beta = 3$. The equation has a simple analytical solution:

$$
u(x,t) = \sin(\pi x)\cos(2\pi t) + \frac{1}{2}\sin(\beta\pi x)\cos(2\beta\pi t) \tag{39}
$$

Table 8: Summary of Hyperparameters in Complex Function Fitting Experiment for Various Models

| Model | Hyperparameters | Model Parameters |
|---|---|---|
| Cheby1KAN | Layer 1: Cheby1KANLayer(1, 7, 8) 
 Layer 2: Cheby1KANLayer(7, 8, 8) 
 Layer 3: Cheby1KANLayer(8, 1, 8) | 639 |
| Cheby2KAN | Layer 1: Cheby2KANLayer(1, 7, 8) 
 Layer 2: Cheby2KANLayer(7, 8, 8) 
 Layer 3: Cheby2KANLayer(8, 1, 8) | 639 |
| PINN | Layer 1: Linear(in=1, out=16), Activation=Tanh 
 Layer 2: Linear(in=16, out=32), Activation=Tanh 
 Layer 3: Linear(in=32, out=1) | 609 |
| AC-PKAN$_s$ | Linear Embedding: Linear(in=1, out=4) 
 Hidden ChebyKAN Layers: $2 \times$ Cheby1KANLayer() 
 Hidden LN Layers: $2 \times$ LayerNorm(features=6) 
 Output Layer: Linear(in=6, out=1) 
 Activations: WaveAct (U and V) | 751 |
| KAN | Layers: $2 \times$ KANLinear (32 neurons, SiLU activation) | 640 |
| rKAN | Layer 1: Linear(in=1, out=16), Activation=JacobiRKAN() 
 Layer 2: Linear(in=16, out=32), Activation=PadeRKAN() 
 Layer 3: Linear(in=32, out=1) | 626 |
| fKAN | Layer 1: Linear(in=1, out=16), Activation=FractionalJacobiNeuralBlock() 
 Layer 2: Linear(in=16, out=32), Activation=FractionalJacobiNeuralBlock() 
 Layer 3: Linear(in=32, out=1) | 615 |
| FastKAN | FastKANLayer 1: 
   RBF 
   SplineLinear(in=8, out=32) 
   Base Linear(in=1, out=32) 
 FastKANLayer 2: 
   RBF 
   SplineLinear(in=256, out=1) 
   Base Linear(in=32, out=1) | 658 |
| FourierKAN | FourierKANLayer 1: NaiveFourierKANLayer() 
 FourierKANLayer 2: NaiveFourierKANLayer() 
 FourierKANLayer 3: NaiveFourierKANLayer() | 685 |

**1D-Reaction PDE.** The one-dimensional reaction problem is a hyperbolic PDE that is commonly used to model chemical reactions. The system has the formulation with periodic boundary conditions as follows:

$$\frac{\partial u}{\partial t} - \rho u(1 - u) = 0, \ \ \forall x \in [0, 2\pi], \ t \in [0, 1]$$
$$\mathtt{IC:}\, u(x, 0) = \exp(-\frac{(x - \pi)^2}{2(\pi/4)^2}), \ \ \mathtt{BC:}\, u(0, t) = u(2\pi, t) \tag{40}$$

where $\rho$ is the reaction coefficient. Here, we set $\rho = 5$. The equation has a simple analytical solution:

$$u_{\mathtt{analytical}} = \frac{h(x) \exp(\rho t)}{h(x) \exp(\rho t) + 1 - h(x)} \tag{41}$$

where $h(x)$ is the function of the initial condition.

**2D Navier–Stokes PDE** The two-dimensional Navier–Stokes equations are given by:

$$\frac{\partial u}{\partial t} + \lambda_1 \left( u\frac{\partial u}{\partial x} + v\frac{\partial u}{\partial y} \right) = -\frac{\partial p}{\partial x} + \lambda_2 \left( \frac{\partial^2 u}{\partial x^2} + \frac{\partial^2 u}{\partial y^2} \right),$$

$$\frac{\partial v}{\partial t} + \lambda_1 \left( u\frac{\partial v}{\partial x} + v\frac{\partial v}{\partial y} \right) = -\frac{\partial p}{\partial y} + \lambda_2 \left( \frac{\partial^2 v}{\partial x^2} + \frac{\partial^2 v}{\partial y^2} \right),$$

(42)

where $u(t, x, y)$ and $v(t, x, y)$ are the $x$- and $y$-components of the velocity field, respectively, and $p(t, x, y)$ is the pressure field. These equations describe the Navier–Stokes flow around a cylinder. The test errors for $v$ are presented in Table 2.

We set the parameters $\lambda_1 = 1$ and $\lambda_2 = 0.01$. Since the system lacks an explicit analytical solution, we utilize the simulated solution provided in Raissi et al. (2019). We focus on the prototypical problem of incompressible flow past a circular cylinder, a scenario known to exhibit rich dynamic behavior and transitions across different regimes of the Reynolds number, defined as Re $= \frac{u_\infty D}{\nu}$. By assuming a dimensionless free-stream velocity $u_\infty = 1$, a cylinder diameter $D = 1$, and a kinematic viscosity $\nu = 0.01$, the system exhibits a periodic steady-state behavior characterized by an asymmetric vortex shedding pattern in the cylinder wake, commonly known as the Kármán vortex street. All experimental settings are the same as in Raissi et al. (2019). For more comprehensive details about this problem, please refer to that work.

**Data Generation:** For all experiments except the 2D Navier–Stokes equation experiment, data points were generated to facilitate the training and testing of the Physics-Informed Neural Network (PINN) within the spatial domain $x \in [0, 1]$ and the temporal domain $t \in [0, 1]$. The data generation process was executed as follows:

- **Grid Creation**: A uniform grid was established using 101 equidistant points in both the spatial ($x$) and temporal ($t$) dimensions, resulting in a total of $101 \times 101 = 10{,}201$ collocation points.

- **Boundary and Initial Conditions**: Boundary points were extracted from the grid to enforce Dirichlet boundary conditions, while the initial condition points were identified at $t = 0$.

- **Tensor Conversion**: The generated data points were converted into PyTorch tensors with floating-point precision and were set to require gradients for automatic differentiation. All experiments were conducted on an NVIDIA A100 GPU with 40 GB of memory.

**Training and Test Sets**: The training and test sets each consist of two distinct groups containing $101 \times 101 = 10{,}201$ collocation points. These points were generated using the data generation method described above.

For the 2D Navier–Stokes equation experiment, the dataset used is detailed as follows:

| Variable | Dimensions | Description |
|---|---|---|
| $X$ (Spatial Coordinates) | (5000, 2) | Contains 5,000 spatial points, each with 2 coordinate values ($x$ and $y$). |
| $t$ (Time Data) | (200, 1) | Contains 200 time steps, each corresponding to a scalar value. |
| $U$ (Velocity Field) | (5000, 2, 200) | Contains 5,000 spatial points, 2 velocity components ($u$ and $v$), and 200 time steps. The velocity data of each point is a function of time. |
| $P$ (Pressure Field) | (5000, 200) | Contains pressure data for 5,000 spatial points and 200 time steps. |

Table 9: Dataset used in the 2D Navier–Stokes equation experiment

**Training and Test Sets**: From the total dataset of 1,000,000 data points ($N \times T = 5{,}000 \times 200$), we randomly selected 2,500 samples for training, which include coordinate positions, time steps, and the corresponding velocity and pressure components. The test set consists of all spatial data at the 100th time step.

**Epochs:**  We trained the models until convergence but did not exceed 50,000 epochs.

**Reproducibility:**  To ensure reproducibility of the experimental results, all random number generators are seeded with a fixed value (seed = 0) across NumPy, Python's `random` module, and PyTorch (both CPU and GPU).

**Running Time**  We present the actual running times (hours:minutes:seconds) for all five PDE experiments in the paper. As shown in Table 10, AC-PKAN demonstrates certain advantages among the KAN model variants, although the running times of all KAN variants are relatively long. This is primarily because the KAN model is relatively new and still in its preliminary stages; although it is theoretically innovative, its engineering implementation remains rudimentary and lacks deeper optimizations. Moreover, while traditional neural networks benefit from well-established optimizers such as Adam and L-BFGS, optimization schemes specifically tailored for KAN have not yet been thoroughly explored. We believe that the performance of AC-PKAN will be further enhanced as the overall optimization strategies for KAN variants improve.

| Model | 1D-Reaction | 1D-Wave | Heterogeneous Problem | Complex Geometry | 2D-NS |
|---|---|---|---|---|---|
| PINN | 00:09:07 | 00:21:14 | 00:23:30 | 00:01:08 | 00:15:20 |
| PINNsFormer | 00:04:09 | 00:44:21 | 14:01:55 | 00:13:31 | 00:58:54 |
| QRes | 00:02:10 | 01:41:34 | 00:20:50 | 00:01:46 | 00:24:39 |
| FLS | 00:01:29 | 01:38:01 | 00:13:38 | 00:01:08 | 00:11:51 |
| Cheby1KAN | 00:12:08 | 03:32:10 | 00:50:45 | 00:03:21 | 04:24:59 |
| Cheby2KAN | 01:06:54 | 05:03:18 | 01:35:40 | 00:03:27 | 05:41:42 |
| AC-PKAN | 00:15:16 | 01:13:01 | 01:13:11 | 00:01:04 | 02:21:40 |
| KINN | 03:04:19 | 25:00:20 | 01:51:44 | 00:14:07 | 14:31:42 |
| rKAN | 01:21:25 | 12:44:16 | 06:21:00 | 00:16:06 | 05:19:04 |
| FastKAN | 05:51:21 | 09:35:51 | 03:37:57 | 00:17:23 | 02:04:42 |
| fKAN | 00:13:09 | 08:20:34 | 00:52:05 | 00:06:22 | 03:01:41 |
| FourierKAN | 01:21:50 | 03:33:46 | 07:40:43 | 00:18:26 | 02:48:50 |

Table 10: Running times (hours:minutes:seconds) for all five PDE experiments

**Hyperparameter Selection:**  The weights used in the external RBA attention are dynamically updated during training using smoothing factor $\eta = 0.001$ and $\beta_w = 0.001$. Different models employed in our experiments have varying hyperparameter configurations tailored to their specific architectures. Table 11 summarizes the hyperparameters and the total number of parameters for each model.

### D.3    PDEs in Complex Engineering Environments Setup Details

In this study, we investigate the performance of AC-PKAN compared with other models in solving complex PDEs characterized by heterogeneous material properties and intricate geometric domains. Specifically, we focus on two distinct difficult environmental PDE problems: a heterogeneous Poisson problem and a Poisson equation defined on a domain with complex geometric conditions. The following sections detail the formulation of the PDEs, data generation processes, model architecture, training regimen, hyperparameter selection, and evaluation methodologies employed in our experiments.

**Heterogeneous Poisson Problem.**  We consider a two-dimensional Poisson equation with spatially varying coefficients to model heterogeneous material properties. The PDE is defined as:

$$\begin{cases} a_1 \Delta u(\boldsymbol{x}) = 16r^2 & \text{for } r < r_0, \\ a_2 \Delta u(\boldsymbol{x}) = 16r^2 & \text{for } r \geq r_0, \\ u(\boldsymbol{x}) = \frac{r^4}{a_2} + r_0^4 \left( \frac{1}{a_1} - \frac{1}{a_2} \right) & \text{on } \partial\Omega, \end{cases} \tag{43}$$

where $r = \|\boldsymbol{x}\|_2$ is the distance from the origin, $a_1 = \frac{1}{15}$ and $a_2 = 1$ are the material coefficients, $r_0 = 0.5$ defines the interface between the two materials, and $\partial\Omega$ represents the boundary

of the square domain $\Omega = [-1, 1]^2$. The boundary condition is a pure Dirichlet condition applied uniformly on all four edges of the square.

**Complex Geometric Poisson Problem.** Additionally, we examine a Poisson equation defined on a domain with complex geometry, specifically a rectangle with four circular exclusions. The PDE is given by:

$$-\Delta u = 0 \quad \text{in } \Omega = \Omega_{\text{rec}} \setminus \bigcup_{i=1}^{4} R_i, \tag{44}$$

where $\Omega_{\text{rec}} = [-0.5, 0.5]^2$ is the rectangular domain and $R_i$ for $i = 1, 2, 3, 4$ are circular regions defined as:

$$R_1 = \left\{ (x, y) : (x - 0.3)^2 + (y - 0.3)^2 \leq 0.1^2 \right\},$$
$$R_2 = \left\{ (x, y) : (x + 0.3)^2 + (y - 0.3)^2 \leq 0.1^2 \right\},$$
$$R_3 = \left\{ (x, y) : (x - 0.3)^2 + (y + 0.3)^2 \leq 0.1^2 \right\},$$
$$R_4 = \left\{ (x, y) : (x + 0.3)^2 + (y + 0.3)^2 \leq 0.1^2 \right\}.$$

The boundary conditions are specified as:

$$u = 0 \quad \text{on } \partial R_i, \quad \forall i = 1, 2, 3, 4, \tag{45}$$
$$u = 1 \quad \text{on } \partial \Omega_{\text{rec}}. \tag{46}$$

**Data Generation:** To train and evaluate the PINNs, we generate collocation points within the defined spatial domains and enforce boundary conditions appropriately.

- **Grid Creation**: For both PDE problems, a uniform grid is established using 100 equidistant points in each spatial dimension, resulting in $101 \times 101 = 10,201$ internal collocation points for the heterogeneous Poisson problem and an analogous number for the complex geometric Poisson problem.
- **Boundary Sampling**:
  - **Heterogeneous Poisson Problem**: Boundary points are extracted from the edges of the square domain $\Omega = [-1, 1]^2$ to impose Dirichlet boundary conditions.
  - **Complex Geometric Poisson Problem**: Boundary points are sampled from both the outer boundary $\partial \Omega_{\text{rec}}$ and the boundaries of the excluded circular regions $\partial R_i$ for $i = 1, 2, 3, 4$.
- **Tensor Conversion**: All collocation and boundary points are converted into PyTorch tensors with floating-point precision and are set to require gradients to facilitate automatic differentiation. The data resides on an NVIDIA A100 GPU with 40GB of memory to expedite computational processes.

The test datasets for both PDE problems mirror the training datasets in terms of spatial discretization, ensuring consistency in the evaluation of the model's generalization capabilities.

**Training Regimen:** Both PDE problems are trained for a total of 50,000 epochs to allow sufficient learning iterations. And the RBA attention mechanism for AC-PKAN is configured with smoothing factors $\eta = 0.001$ and $\beta_w = 0.001$.

**Reproducibility:** To ensure the reproducibility of our experimental results, all random number generators are seeded with a fixed value (seed = 0) across NumPy, Python's `random` module, and PyTorch (both CPU and GPU). This deterministic setup guarantees consistent initialization and training trajectories across multiple runs.

**Hyperparameter Selection:** Table 12 summarizes the hyperparameters and the total number of parameters for each model.

## E RESULTS DETAILS AND VISUALIZATIONS.

Firstly, in the context of the 1D-Wave experiment, we plotted the values of $\log\left(\lambda_{IC,BC}^{\text{GRA}}\right)$ over epochs in Figure 6 as the values of $\lambda_{IC,BC}^{\text{GRA}}$ over epochs is presented in Figure 3c.

Then we illustrate the fitting results of nine models for complex functions in Figure 7. Additionally, we present the plots of ground truth solutions, neural network predictions, and absolute errors for all evaluations conducted in the five PDE-solving experiments. The results for the 1D-Reaction, 1D-Wave, 2D Navier-Stokes, Heterogeneous Poisson Problem, and Complex Geometric Poisson Problem are displayed in Figures 10, 8, 9, and 11, respectively.

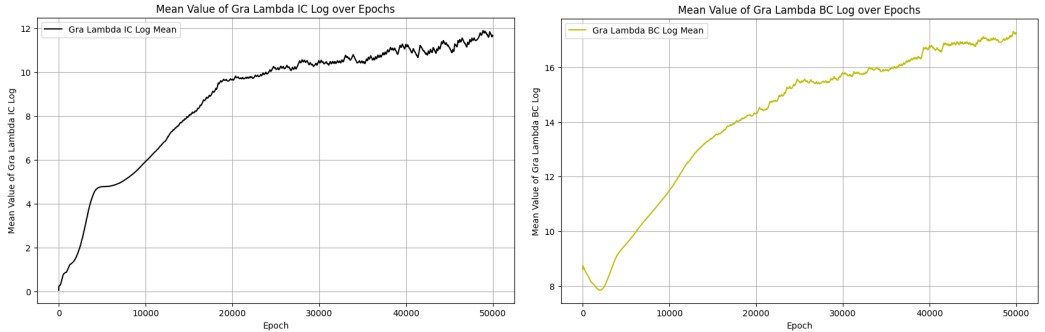

Figure 6: Mean values of GRA weights after logarithmic transformation over epochs for the 1D-Wave experiment.

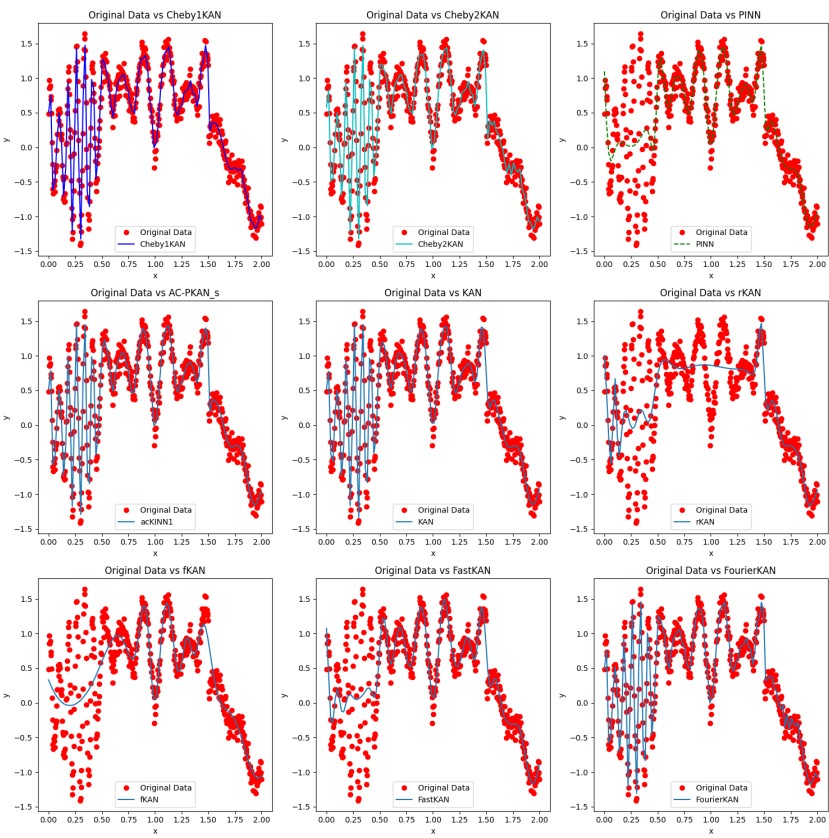

Figure 7: Illustration of 9 Various Models for Complex Function Fitting

Table 11: Summary of Hyperparameters in PINN Failure Modes Experiment for Various Models

| Model | Hyperparameters | Model Parameters |
|---|---|---|
| AC-PKAN | Linear Embedding: $2 \rightarrow 64$
Hidden ChebyKAN Layers: $3 \times$ Cheby1KANLayer (degree=8)
Hidden LN Layers: $3 \times$ LayerNorm (128)
Output Layer: $128 \rightarrow 1$
Activations: WaveAct | 460,101 |
| QRes | Input Layer: QRes_block ($2 \rightarrow 256$, Sigmoid)
Hidden Layers: $3 \times$ QRes_block ($256 \rightarrow 256$, Sigmoid)
Output Layer: $256 \rightarrow 1$ | 396,545 |
| FastKAN | Layer 1: FastKANLayer (RBF, SplineLinear $16 \rightarrow 8500$, Base Linear $2 \rightarrow 8500$)
Layer 2: FastKANLayer (RBF, SplineLinear $68,000 \rightarrow 1$, Base Linear $8500 \rightarrow 1$) | 246,518[*] |
| KAN | Layers: $2 \times$ KANLinear (9000 neurons, SiLU activation) | 270,000[*] |
| PINNs | Sequential Layers:
$2 \rightarrow 512$ (Linear, Tanh)
$512 \rightarrow 512$ (Linear, Tanh)
$512 \rightarrow 512$ (Linear, Tanh)
$512 \rightarrow 1$ (Linear) | 527,361 |
| FourierKAN | NaiveFourierKANLayer 1: $2 \rightarrow 32$, Degree=8
NaiveFourierKANLayer 2: $32 \rightarrow 128$, Degree=8
NaiveFourierKANLayer 3: $128 \rightarrow 128$, Degree=8
NaiveFourierKANLayer 4: $128 \rightarrow 32$, Degree=8
NaiveFourierKANLayer 5: $32 \rightarrow 1$, Degree=8 | 395,073 |
| Cheby1KAN | Cheby1KANLayer 1: $2 \rightarrow 32$, Degree=8
Cheby1KANLayer 2: $32 \rightarrow 128$, Degree=8
Cheby1KANLayer 3: $128 \rightarrow 256$, Degree=8
Cheby1KANLayer 4: $256 \rightarrow 32$, Degree=8
Cheby1KANLayer 5: $32 \rightarrow 1$, Degree=8 | 406,368 |
| Cheby2KAN | Cheby2KANLayer 1: $2 \rightarrow 32$, Degree=8
Cheby2KANLayer 2: $32 \rightarrow 128$, Degree=8
Cheby2KANLayer 3: $128 \rightarrow 256$, Degree=8
Cheby2KANLayer 4: $256 \rightarrow 32$, Degree=8
Cheby2KANLayer 5: $32 \rightarrow 1$, Degree=8 | 406,368 |
| fKAN | Sequential Layers:
$2 \rightarrow 256$ (Linear, fJNB(3))
$256 \rightarrow 512$ (Linear, fJNB(6))
$512 \rightarrow 512$ (Linear, fJNB(3))
$512 \rightarrow 128$ (Linear, fJNB(6))
$128 \rightarrow 1$ (Linear) | 460,813 |
| rKAN | Sequential Layers:
$2 \rightarrow 256$ (Linear, JacobiRKAN(3))
$256 \rightarrow 512$ (Linear, PadeRKAN[2/6])
$512 \rightarrow 512$ (Linear, JacobiRKAN(6))
$512 \rightarrow 128$ (Linear, PadeRKAN[2/6])
$128 \rightarrow 1$ (Linear) | 460,835 |
| FLS | Sequential Layers:
$2 \rightarrow 512$ (Linear, SinAct)
$512 \rightarrow 512$ (Linear, Tanh)
$512 \rightarrow 512$ (Linear, Tanh)
$512 \rightarrow 1$ (Linear) | 527,361 |
| PINNsformer | Parameters: d_out=1, d_hidden=512, d_model=32, N=1, heads=2 | 453,561 |

[*] This reaches the GPU memory limit, and increasing the number of parameters further would cause an out-of-memory error.

Table 12: Summary of Hyperparameters in Complex Engineering Environmental PDEs for Various Models

| Model | Hyperparameters | Model Parameters |
|---|---|---|
| AC-PKAN | Linear Embedding: in=2, out=32
ChebyKAN Layers: 4 layers, degree=8
LN Layers: 4 layers, features=64
Output Layer: in=64, out=1
Activation: WaveAct | 152,357 |
| QRes | Input Layer: in=2, out=128
Hidden Layers: 5 QRes blocks, units=128
Output Layer: in=128, out=1
Activation: Sigmoid | 166,017 |
| PINN | Layer 1: $2 \rightarrow 256$, Activation=Tanh
Layer 2: $256 \rightarrow 512$, Activation=Tanh
Layer 3: $512 \rightarrow 128$, Activation=Tanh
Layer 4: $128 \rightarrow 1$ | 198,145 |
| PINNsformer | d_out=1
d_hidden=128
d_model=8
N=1
heads=2 | 158,721 |
| FLS | Layer 1: $2 \rightarrow 256$, Activation=SinAct
Layer 2: $256 \rightarrow 256$, Activation=Tanh
Layer 3: $256 \rightarrow 256$, Activation=Tanh
Layer 4: $256 \rightarrow 1$ | 132,609 |
| Cheby1KAN | Layer 1: $2 \rightarrow 32$, Degree=8
Layer 2: $32 \rightarrow 128$, Degree=8
Layer 3: $128 \rightarrow 64$, Degree=8
Layer 4: $64 \rightarrow 32$, Degree=8
Layer 5: $32 \rightarrow 1$, Degree=8 | 129,888 |
| Cheby2KAN | Layer 1: $2 \rightarrow 32$, Degree=8
Layer 2: $32 \rightarrow 128$, Degree=8
Layer 3: $128 \rightarrow 64$, Degree=8
Layer 4: $64 \rightarrow 32$, Degree=8
Layer 5: $32 \rightarrow 1$, Degree=8 | 129,888 |
| KAN[*] | Layers: $2 \times$ KANLinear
Neurons: 9000
Activation: SiLU | 60,000[*] |
| rKAN | Layer 1: $2 \rightarrow 256$, Activation=JacobiRKAN(3)
Layer 2: $256 \rightarrow 256$, Activation=PadeRKAN[2/6]
Layer 3: $256 \rightarrow 256$, Activation=JacobiRKAN(6)
Layer 4: $256 \rightarrow 128$, Activation=PadeRKAN[2/6]
Layer 5: $128 \rightarrow 1$ | 165,411 |
| FastKAN[*] | FastKANLayer 1: RBF, SplineLinear $16 \rightarrow 2600$, Base Linear $2 \rightarrow 2600$
FastKANLayer 2: RBF, SplineLinear $20800 \rightarrow 1$, Base Linear $2600 \rightarrow 1$ | 75,418[*] |
| fKAN | Layer 1: $2 \rightarrow 256$, Activation=fJNB(3)
Layer 2: $256 \rightarrow 512$, Activation=fJNB(6)
Layer 3: $512 \rightarrow 512$, Activation=fJNB(3)
Layer 4: $512 \rightarrow 128$, Activation=fJNB(6)
Layer 5: $128 \rightarrow 1$ | 132,618 |
| FourierKAN | Layer 1: $2 \rightarrow 32$
Layer 2: $32 \rightarrow 64$
Layer 3: $64 \rightarrow 64$
Layer 4: $64 \rightarrow 64$
Layer 5: $64 \rightarrow 1$
Degree=8 | 166,113 |

[*] This reaches the GPU memory limit, and increasing the number of parameters further would cause an out-of-memory error.

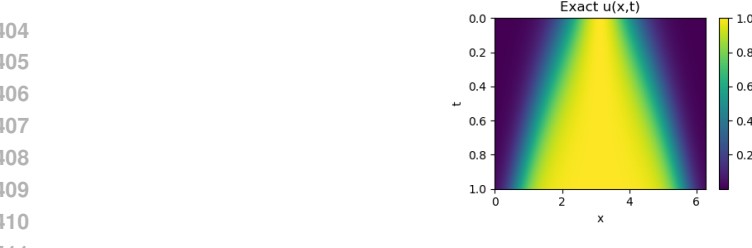

(a) Ground Truth Solution for the 1D-Reaction Equation

(b) From left to right, the first, third, and fifth rows display the predictions of the AC-PKAN, Cheby1KAN, Cheby2KAN, and FastKAN models; the PINNs, QRes, rKAN, and fKAN models; and the PINNsformer, FLS, FourierKAN, and KINN models, respectively. The second, fourth, and sixth rows present their corresponding absolute errors.

Figure 8: Comparison of the ground truth solution for the 1D-Reaction equation with predictions and error maps from various models. The top image illustrates the ground truth, while the subsequent 24 images display the predictions and their respective errors organized in a 6x4 grid, providing a comprehensive overview of each model's performance.

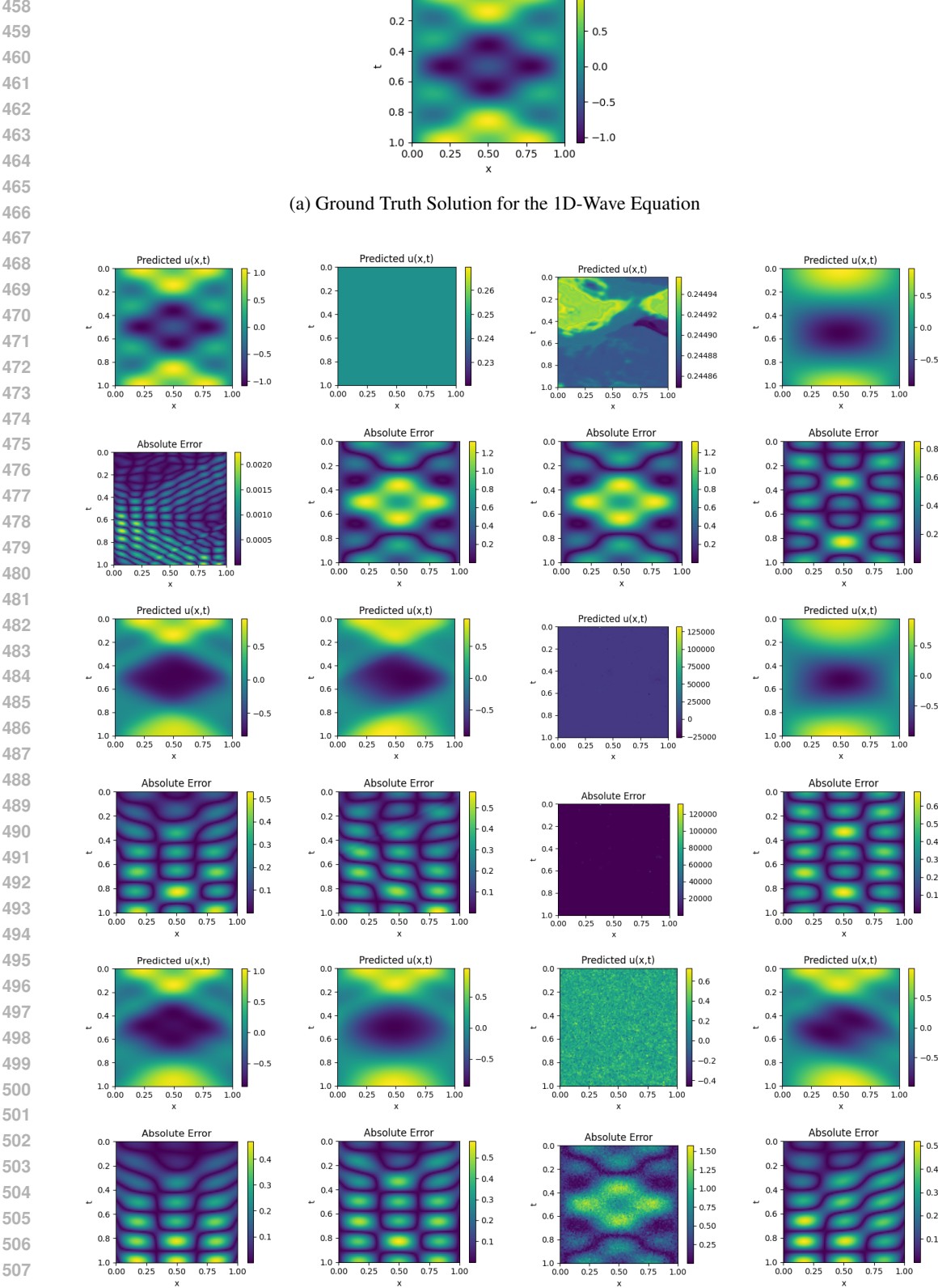

(a) Ground Truth Solution for the 1D-Wave Equation

(b) From left to right, the first, third, and fifth rows display the predictions of the AC-PKAN, Cheby1KAN, Cheby2KAN, and FastKAN models; the PINNs, QRes, rKAN, and fKAN models; and the PINNsformer, FLS, FourierKAN, and KINN models, respectively. The second, fourth, and sixth rows present their corresponding absolute errors.

Figure 9: Comparison of the ground truth solution for the 1D-Wave equation with predictions and error maps from various models.

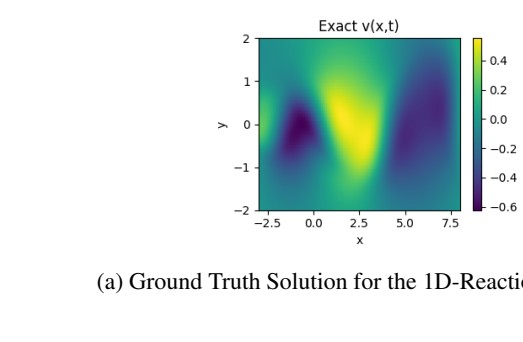

(a) Ground Truth Solution for the 1D-Reaction Equation

(b) From left to right, the first, third, and fifth rows display the predictions of the AC-PKAN, Cheby1KAN, Cheby2KAN, and FastKAN models; the PINNs, QRes, and fKAN models; and the PINNsformer, FLS, FourierKAN, and KINN models, respectively. The second, fourth, and sixth rows present their corresponding absolute errors.

Figure 10: Comparison of the ground truth solution for the 2D Navier-Stokes equation with predictions and error maps from various models. The top image illustrates the ground truth, while the subsequent 22 images display the predictions and their respective errors organized in a 6x4 grid, providing a comprehensive overview of each model's performance.



(a) Ground Truth Solution for the Heterogeneous Possion equation

(b) From left to right, the first, third, and fifth rows display the predictions of the AC-PKAN, Cheby1KAN, Cheby2KAN, and FastKAN models; the PINNs, QRes, rKAN, and fKAN models; and the PINNsformer, FLS, FourierKAN, and KINN models, respectively. The second, fourth, and sixth rows present their corresponding absolute errors.

Figure 11: Comparison of the ground truth solution for the Heterogeneous Possion equation problem with predictions and error maps from various models.

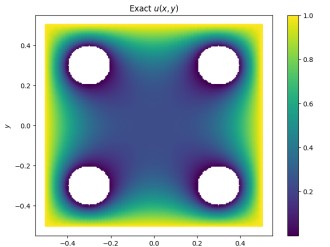

(a) Ground Truth Solution for the Complex Geometry Possion equation

(b) From left to right, the first, third, and fifth rows display the predictions of the AC-PKAN, Cheby1KAN, Cheby2KAN, and FastKAN models; the PINNs, QRes, rKAN, and fKAN models; and the PINNsformer, FLS, FourierKAN, and KINN models, respectively. The second, fourth, and sixth rows present their corresponding absolute errors.

Figure 12: Comparison of the ground truth solution for the Complex Geometry Possion equation problem with predictions and error maps from various models.

