# OpenReview forum: "AC-PKAN: Attention-Enhanced and Chebyshev Polynomial-Based Physics-Informed Kolmogorov–Arnold Networks"
_ICLR.cc/2025/Conference — Submitted to ICLR 2025_

### Official Review · Reviewer_w3dV · 2024-10-20

**Soundness:** 2
**Presentation:** 3
**Contribution:** 2
**Rating:** 5
**Confidence:** 4

**Summary:**

This paper introduces AC-PKAN to fit functions and solve PDEs. The proposed AC-PKAN method employs a Residual Gradient Attention (RGA) mechanism to dynamically adjusts loss term weights, wavelet-activated MLPs with learnable parameters, and Chebyshev polynomial
Based KANs to improve training efficiency and prediction accuracy. The paper is well-written. The method is effective compared to existing methods on five benchmarks.

**Strengths:**

1. The method AC-RKAN is original. Ablation studies show the necessity of each module of AC-RKAN.
2. The paper is well-written. The theoretical insight is interesting and the experiments are rich, demonstrating the effective of the method.
3. The AC-RKAN method can make a timely contribution to the community of physics-informed machine learning.

**Weaknesses:**

1.The experiments and the comparisons are not challenging. For the 1D-wave case, the author claimed PINNsFormer has a relative l2 norm 0.32 in Table 2, but Wang 2022 (fig.6 of their paper) has trained PINN for this case to achieve a relative l2 norm 1.7e-3, which is the same order to AC-KAN. The other PDEs are also simple 2d Poisson eq. Although with Heterogeneous Problem and Complex Geometry, the solution is smooth, and are easy to solve by simple traditional numerical methods such as finite element methods. Challenging PINN problems Wang 2023 such as the Kuramoto–Sivashinsky equation with chaotic behavior，Lid-driven cavity flow (Re=3200)， Navier–Stokes flow in a torus or around a cylinder, are not reported in the paper.

[a]Wang S, Yu X, Perdikaris P. When and why PINNs fail to train: A neural tangent kernel perspective[J]. Journal of Computational Physics, 2022, 449: 110768.
[b]Wang S, Sankaran S, Wang H, et al. An expert's guide to training physics-informed neural networks[J]. arXiv preprint arXiv:2308.08468, 2023.

2.The method introduction is expatiatory, from page3-7. A concentration of the method introduction is recommended and more theoretical insights and experiments can be discussed in the main text.

**Questions:**

See in the weakness

---

> ### Author Response · Authors · 2024-11-18
> **Rebuttal to Reviewer w3dV**
>
> We thank the reviewer for the thoughtful comments.
>
> **Response to W1**
>
> We mildly disagree with the assertion that "The experiments and the comparisons are not challenging." The experiments we selected are also employed in other studies [1–4]. You mentioned in [5] that for the 1D-wave case, a relative $L^2 $ norm of $1.7 \times 10^{-3} $ was achieved, which is of the same order as that of AC-KAN. However, this result was obtained using additional external learning strategies. In our comparisons with different baselines, we did not utilize extra techniques, opting instead for the simplest possible designs to highlight the intrinsic performance of the models through experiment. Nevertheless, to strengthen our statement, we have added an experiment on Navier–Stokes flow around a cylinder (see **2D Navier–Stokes PDE Experiment**). Moreover, we will discuss the paper [5-6]  in our revised paper.
>
> [1] Zhao, Z., Ding, X., Prakash, B. A. "PINNsformer: A transformer-based framework for physics-informed neural networks." *arXiv preprint* arXiv:2307.11833, 2023.
>
> [2] Wu, H., Luo, H., Ma, Y., et al. "RoPINN: Region Optimized Physics-Informed Neural Networks." *arXiv preprint* arXiv:2405.14369, 2024.
>
> [3] Wang, Y., Sun, J., Bai, J., et al. "Kolmogorov Arnold Informed Neural Network: A physics-informed deep learning framework for solving PDEs based on Kolmogorov Arnold Networks." *arXiv preprint* arXiv:2406.11045, 2024.
>
> [4] Hao, Z., Yao, J., Su, C., et al. "Pinnacle: A comprehensive benchmark of physics-informed neural networks for solving PDEs." *arXiv preprint* arXiv:2306.08827, 2023.
>
> [5] Wang, S., Yu, X., Perdikaris, P. "When and why PINNs fail to train: A neural tangent kernel perspective." *Journal of Computational Physics*, 2022, 449: 110768.
>
> [6]Wang S, Sankaran S, Wang H, et al. An expert's guide to training physics-informed neural networks[J]. arXiv preprint arXiv:2308.08468, 2023.
>
> **Response to W2**
>
> We appreciate your constructive feedback and agree with your suggestion. We will revise our manuscript to streamline the method introduction, incorporate the newly conducted Navier–Stokes flow around a cylinder experiment into the main text, and provide a more in-depth discussion of theoretical insights and experimental results.

---

> > ### Comment · Reviewer_w3dV · 2024-11-20
> >
> > Thanks for the reply. I still have some doubts on the experiments.
> >
> > 1. The "external learning strategies" in above [5] is a simple adaptive weights based on NTK matrix. You mentioned AC-PKAN can be integrated with NTK method to achieve better performance (Line 450). Are they two the same? If is, then for the 1D-wave case, the result in above [5] is relative l2 norm 1.7e-3  and in your report Table 4 gives "PINNs+NTK" rRMSE 0.1489, the equation difference is beta=4 in [5] and beta=3 in your work (Line 952-965). Why they differs so much? If not, can this adaptive weights in [5] be integrated in your method by replacing the adaptive reweighting strategy in Line 191?
> >
> > 2. I appreciate the result for 2D Navier-Stokes equation. The rRMSE results are in a range [1.3,3.7] except for some nan. I wonder whether these methods solves 2D NS eq. well since the rRMSE is relatively large. The rRMSE improvement from PINN to AC-PKAN is 3.49 to 2.44, which is rather limited I think. Can you report more details on this problem?

---

> ### Author Response · Authors · 2024-11-20
>
> Thanks for your patient and timely reply!
>
> **Response to 1**
>
> These two NTK methods are consistent. In fact, to establish an effective comparison, our experimental setup in this section is identical to that of Table 2 in [1]. Furthermore, our results align with the experimental findings presented in Table 2 of [1]. For detailed information, please refer to our code (https://anonymous.4open.science/r/1d_wave_pinn_ntk-188A). However, it is a surprise to find that even slight modifications to the PDE parameters caused significant changes in the performance of PINN when using the NTK method. Specifically, the PDE's rRMSE between β = 4 and β = 3 differed by two orders of magnitude. The underlying reasons for this phenomenon warrant further investigation, as it suggests that the NTK method may still has certain limitations. Enhancing the robustness of the NTK method under varying PDE parameter conditions is a potential avenue for future research, although it is beyond the scope of our current study.
>
> **Response to 2**
>
> First, although our AC-PKAN’s improvement in 2D NS eq may appear limited, it is nevertheless a meaningful advancement. In our study, the primary aim is not to demonstrate that AC-PKAN thoroughly surpasses other PINN models, but rather to offer novel perspectives and alternatives through the application of KAN while achieving desirable accuracy in solving PDE systems. Furthermore, AC-PKAN represents a further—albeit still preliminary—exploratory endeavor building upon the relatively nascent development of KAN. While KAN still has performance aspects awaiting enhancement, it has been widely recognized as a promising new direction for PINNs [7].
>
> To provide details on the two-dimensional Navier–Stokes equations, we set the parameters $\lambda_1 = 1$ and $\lambda_2 = 0.01$. Since the system lacks an explicit analytical solution, we utilize the simulated solution provided in [8]. We focus on the prototypical problem of incompressible flow past a circular cylinder—a scenario known to exhibit rich dynamic behavior and transitions across different regimes of the Reynolds number, defined as $\text{Re} = u_\infty D / \nu$. By assuming a dimensionless free-stream velocity $u_\infty = 1$, a cylinder diameter $D = 1$, and a kinematic viscosity $\nu = 0.01$, the system exhibits a periodic steady-state behavior characterized by an asymmetrical vortex shedding pattern in the cylinder wake, commonly known as the Kármán vortex street. For more comprehensive details about this problem, please refer to Section 4.1.1, "Example (Navier–Stokes equation)," in [8].
>
> [7] Toscano J D, Oommen V, Varghese A J, et al. From PINNs to PIKANs: Recent Advances in Physics-Informed Machine Learning[J]. arXiv preprint arXiv:2410.13228, 2024.
>
> [8] Raissi, Maziar, Paris Perdikaris, and George E. Karniadakis. "Physics-informed neural networks: A deep learning framework for solving forward and inverse problems involving nonlinear partial differential equations." Journal of Computational physics 378 (2019): 686-707.

---

> ### Author Response · Authors · 2024-12-02
>
> We hope this message finds you well.
>
> As the deadline for final evaluations is tomorrow, we would greatly appreciate any insights or suggestions you might have to our previous response. We would also be grateful for any consideration towards improving our current score based on potential revisions.
>
> Thank you very much for your time and understanding.

---

### Official Review · Reviewer_i8XJ · 2024-11-03

**Soundness:** 3
**Presentation:** 3
**Contribution:** 3
**Rating:** 6
**Confidence:** 4

**Summary:**

The paper introduces AC-PKAN, an advanced Physics-Informed Neural Networks (PINNs) framework that integrates Residual Gradient Attention (RGA) and point-wise Chebyshev polynomial-based (KANsCheby1KAN) to address the limitations of traditional PINNs in solving complex partial differential equations. Experimental results demonstrate that AC-PKAN consistently outperforms or matches state-of-the-art models, establishing it as a highly effective tool for engineering applications

**Strengths:**

1. This article conducts quite a few experiments to demonstrate the method's effectiveness compared to other KAN and MLP.
2. This paper's presentation is very clear, and the main idea and proposed optimization are easily understood.

**Weaknesses:**

1. In my opinion, RGA and Cheby1KAN seem to be separated from each other, and there is a feeling that two methods were found to sew the optimized PINN and KAN respectively. It is necessary to explain whether Cheby1KAN is more effective for your proposed RGA compared with other network layers through theory.
2. In the experiment part, only the comparison of different KAN and MLP used by PINN, more comparisons should be added such as FNO-related methods.

**Questions:**

1. Is the dataset trained in different batches, if there is, how can we calculate the maximum Loss of the entire training data in Equation 4.

---

> ### Author Response · Authors · 2024-11-18
> **Rebuttal to Reviewer i8XJ**
>
> We thank the reviewer for the comments.
>
> **Response to W1**
>
> As shown in **Ablation Study 2**, our ablation experiments demonstrate that AC-PKAN leverages our proposed RGA more effectively than other frameworks.
>
> **Response to W2**
>
> FNO-related methods and PINNs belong to entirely different categories of research. FNO falls under another major direction of AI4PDEs—operator learning, which aims to learn a mapping operator rather than solving a specific partial differential equation.
>
> FNO is a purely data-driven deep learning model without incorporation of physical constraints, whereas PINNs are strongly physics-constrained models with minimal reliance on data. Their learning objectives and implementation approaches are fundamentally different. As a result, it is standard practice[1-5] not to compare FNO-related methods with PINNs in PINN-related studies, nor to use PINNs as baselines in operator learning research.
>
> [1] Zhao, Z., Ding, X., Prakash, B. A. "PINNsformer: A transformer-based framework for physics-informed neural networks." arXiv preprint arXiv:2307.11833, 2023.
>
> [2] Wu, H., Luo, H., Ma, Y., et al. "RoPINN: Region Optimized Physics-Informed Neural Networks." arXiv preprint arXiv:2405.14369, 2024.
>
> [3] Baez A, Zhang W, Ma Z, et al. Guaranteeing Conservation Laws with Projection in Physics-Informed Neural Networks[J]. arXiv preprint arXiv:2410.17445, 2024.
>
> [4] Bonev B, Kurth T, Hundt C, et al. Spherical fourier neural operators: Learning stable dynamics on the sphere[C]//International conference on machine learning. PMLR, 2023: 2806-2823.
>
> [5] Tran A, Mathews A, Xie L, et al. Factorized fourier neural operators[J]. arXiv preprint arXiv:2111.13802, 2021.
>
> **Response to Q1**
>
> No, it isn’t. Given that the dataset is extremely small—if it exists at all—it is unnecessary to train it using different batches.
>
> PINNs are strongly physics-constrained models that embed PDE constraints directly into the loss function for optimization, requiring minimal or no data. This characteristic represents a fundamental distinction from operator learning. For instance, in our five PDE-solving experiments, no additional data was used except for the Navier–Stokes equation, which required approximately 24.08 MB of additional training data. In all other cases, no external data constraints were applied. Even when data is used, the dataset size is extremely small, and the entire dataset is utilized in every training iteration of AC-PKAN, eliminating the need to divide it into different batches.

---

> > ### Comment · Reviewer_i8XJ · 2024-11-25
> > **Official Comment by Reviewer i8XJ**
> >
> > Thank you for your detailed reply. I have a few additional questions and concerns:
> >
> > > As shown in Ablation Study 2, our ablation experiments demonstrate that AC-PKAN leverages our proposed RGA more effectively than other frameworks.
> >
> > Regarding Ablation Study 2 in Table 5, it demonstrates that the inclusion of RGA in AC-PKAN provides advantages. However, it does not offer a direct comparison with other methods. My primary question is whether there is any theoretical connection between RGA and Cheby1KAN or if their designs originate from addressing the same underlying problem. If the intent is solely to improve performance, the approach may appear as a module stacking rather than a principled solution. Could you clarify your insight to use RGA and Cheby1KAN together further?
> >
> > > For instance, in our five PDE-solving experiments, no additional data was used except for the Navier–Stokes equation, which required approximately 24.08 MB of additional training data.
> >
> > I noticed that the paper does not provide detailed statistics about the datasets used. Could you include a table that specifies the size of the training and test sets, as well as the resolution of inputs and outputs for each dataset? Relying on file sizes as a comparison metric is not rigorous and lacks clarity.

---

> ### Author Response · Authors · 2024-11-24
>
> Dear Reviewer i8XJ,
>
> We appreciate your time for reviewing, and we really want to have a further discussion with you to see if our response solves the concerns. We have addressed all the thoughtful questions raised by the reviewer (eg, technical descriptions and contributions), and we hope that our work’s impact and results are better highlighted with our responses. It would be great if the reviewer can kindly check our responses and provide feedback with further questions/concerns (if any). We would be more than happy to address them.
>
> Thank you!
>
> Best wishes,
>
> Authors

---

> ### Author Response · Authors · 2024-11-25
>
> Thank you for your thoughtful reply!
>
> **Response 1**
>
> Our RGA mechanism is composed of Gradient-Related Attention (GRA) and Residual-Based Attention (RBA).
>
> GRA is employed to alleviate the stiffness problem in the gradient flow dynamics of the PDE loss term in PINNs, as discussed in detail in [6]. Operations within the Cheby1KAN layer exacerbate this stiffness due to:
>
> •	Higher-Order Polynomials: Computing PDE residuals requires high-order derivatives, which necessitates higher-degree Chebyshev polynomial terms, leading to large coefficients and derivative magnitudes. This increases the Hessian's maximum eigenvalue, intensifying gradient flow stiffness. Moreover, higher-order polynomial terms are particularly prone to generating rapidly fluctuating numerical values, especially as the input $x$ approaches the boundaries of the interval. In such scenarios, the amplified contributions of these terms lead to instability in parameter updates along the affected directions.
>
> •	Nonlinear Operations: The nonlinear functions $\cos$ and $\arccos$ have gradients can vary significantly in certain regions. The gradient of $\cos(x)$ is $-\sin(x)$, which approaches zero as x→π/2 (gradient vanishing), while the gradient of $\arccos(x)$ is $-\frac{1}{\sqrt{1 - x^2}}$, tending to infinity as x→±1 (gradient explosion).
>
> •	Increased Nonlinearity with Higher Degrees: Higher-degree Chebyshev polynomials increase network nonlinearity, leading to more imbalanced loss gradients and exacerbated dynamic stiffness.
>
> RBA, a lightweight dynamic weighting method based on pointwise residuals, is well-suited for computations within the Cheby1KAN layer. While Cheby1KAN captures complex physical features and excels in regions with rapid changes, its complexity can hinder convergence in certain areas. RBA enhances fitting capability by amplifying weights where residuals are high, dynamically balancing residuals across regions and improving training stability. This complements Cheby1KAN's strengths in handling strong nonlinearity and complex distributions, such as high-frequency and singular solutions. Furthermore, RBA's weighting mechanism forms a synchronously adjusted feedback loop with Cheby1KAN, mitigating numerical issues in optimization and enhancing global convergence efficiency.
>
> A rigorous mathematical derivation of the connection between GRA and Cheby1KAN is a future research direction but beyond the scope of our current study.
>
> [6] Wang S, Teng Y, Perdikaris P. Understanding and mitigating gradient flow pathologies in physics-informed neural networks. SIAM Journal on Scientific Computing, 2021, 43(5): A3055–A3081.
>
> **Response 2**
>
> Firstly, in the four PDE experiments presented in our original manuscript, we did not utilize any external datasets; the data were self-generated. Specifically, we sampled points within the plane and computed the corresponding physical constraint PDE losses, as well as initial or boundary condition losses. For detailed information, please refer to lines 978–994 in the original manuscript(lines 1100-1134 in the revised pdf). We would like to further supplement this section as follows:
>
> *Training and Test Sets*: The training and test sets each consist of two distinct groups containing $101 \times 101 = 10,201$ collocation points. These points are generated using the data generation method described earlier.
>
> Secondly, for the 2D Navier–Stokes equation experiment added during the rebuttal phase, the dataset used are explained as follows:
>
> |**Variable**|**Dimensions**|**Description**|
> |-|-|-|
> |$X$ (Spatial Coordinates)|$(5000, 2)$|Contains 5,000 spatial points, each with 2 coordinate values ($x$ and $y$).|
> |$t$ (Time Data)|$(200, 1)$|Contains 200 time steps, each corresponding to a scalar value.|
> |$U$ (Velocity Field)|$(5000, 2, 200)$|Contains 5,000 spatial points, 2 velocity components ($u$ and $v$), and 200 time steps. The velocity data of each point is a function of time.|
> |$P$ (Pressure Field)|$(5000, 200)$|Contains pressure data for 5,000 spatial points and 200 time steps.|
>
> *Training and Test Sets*: From the total dataset of 1,000,000 data points ($N \times T = 5,000 \times 200$), we randomly selected 2,500 samples for training, which include coordinate positions, time, and corresponding velocity and pressure components. The test set consists of all spatial data at the 100th time step.

---

> > ### Comment · Reviewer_i8XJ · 2024-11-27
> > **Official Comment by Reviewer i8XJ**
> >
> > Thanks for your reply, I think you need to explain the specific details of the dataset in your paper.
> > Overall, I think there are no obvious problems and will raise my score to 6.

---

### Official Review · Reviewer_mYon · 2024-11-03

**Soundness:** 2
**Presentation:** 4
**Contribution:** 2
**Rating:** 6
**Confidence:** 4

**Summary:**

The paper introduces several new mechanisms to address the failure of PINNs and difficulties of PINNs to extend to higher complexities.The paper proposes the residual gradient attention mechanism to dynamically adjust the loss coefficients of residuals and gradient norms.   It also improves the Kolmogorov-Arnold Networks with enhanced attention and a learnable Wavelet function. The paper showcases that AC-PKAN improves over KANN and PINN on complex function interpolation. It also achieves comparable and superior results to SOTA PINN variants in solving PDEs. The authors show the effectiveness of AC-PKAN components by a series of ablation studies.

**Strengths:**

(1) The paper is written with a logical flow and clear exposition, and the authors provide several algorithm pseudocodes that make the algorithm easy to understand.

(2) The empirical performance of AC-PKAN is promising, demonstrating improvements across a range of benchmark tests and scenarios where traditional approaches like PINNs and KANs face challenges.

(3) The authors have proposed several novel modifications to both PINNs and KANs,  to addresses several limitations inherent to PINNs and KANs, such as scalability with respect to equation complexity, convergence stability, and generalizability across various domains.

(4) The authors provide the mean values of GRA weights and RBA weights over epochs in the experiment section. The increase of the GRA weights and significant difference between GRA and RBA justifies the authors' claim that balanced attention will resolve the gradient stiffness problem.

(5) The Loss Landscape Analysis, illustrations of fitting complex functions and plots of loss of PDE fitting are useful to understand how AC-PKAN performs in various setting.

(6) The authors provide a detailed experiment setting and model parameter analysis, allowing for replicating the paper findings.

**Weaknesses:**

(1) While the memory is discussed within the appendix, the paper fails to discuss the running time of each algorithm to fit the PDEs and complex functions in the experiments. It’s unclear to the readers how much different architectures such as KAN, Cheby2KAN, MLP and transformer trades off time with accuracy and efficiency.

(2) The ablation study only discusses the drop of algorithm effectiveness by removing the key components of the AC-PKAN. However, many of the design choices can be adapted to other algorithms, for example, the RGA/RBA attention updates can work with MLP PINNs and First-kind Chebyshev KAN Layer could also work seamlessly with the PINN objective. Without analysis on each individual component, it is hard to see how each component could improve the algorithm performance.

(3) Proposition 1 is a bit vague, it is hard to know what "positive integer N" means in the theorem without having to look at the detailed proofs in the appendix.

(4) Moreoever, while the authors claim that AC-PKAN is able to approximate the PDEs with higher orders and higher dimensionality, there is lack of experiment showing such effect.

(5) For the total loss under the RGA mechanism in Equation 8. The authors state that they would set the scaling factors to 1. However, the scaling factor could be set to larger number, and the authors could remove the logarithmic transformation in the loss function. It may be better to demonstrate the effect of scales vesus setting log transformation to justify the design choices.

**Questions:**

(1) In Algorithm 1, the compute gradients and modify parameters uses single SGD, could we apply the RGA Mechanism with other gradient update techniques such as Adam or AdamW?

(2) PINN suffers from the curse of dimensionality of high-dimensional spaces, does AC-PKAN resolve such issues on multi-dimensional PDE equations? The current experiment focuses all on 1-D PDEs and complex equations.

(3) Proposition 1 states that the attention mechanism and wavelet activations in AC-PKAN ensure the output function has non-zero derivatives of all orders. Can a simpler architecture, like an MLP with sinusoidal activations, also guarantee non-zero derivatives in its output?

(4) The ablation study reveals that removing the linear layer has the most significant impact on AC-PKAN’s performance. Could the authors clarify why this occurs? Why does the linear layer play such a crucial role in AC-PKAN, while other PINN methods do not require similar linear projections?

---

> ### Author Response · Authors · 2024-11-18
> **Rebuttal to Reviewer mYon**
>
> We thank the reviewer for the thoughtful comments. Your suggestions are very important for improving our paper.
>
> **Response to W1**
>
> We present the actual running time data(hours: minutes: seconds) for all five PDE experiments in the paper, including the Navier–Stokes equation that we additionally conducted.
>
> |Model|1D-Reaction|1D-Wave|Heterogeneous Problem|Complex Geometry|Navier–Stokes|
> |-|-|-|-|-|-|
> |PINN|00:09:07|00:21:14|00:23:30|00:01:08|00:15:20|
> |PINNsformer|00:04:09|00:44:21|14:01:55|00:13:31|00:58:54|
> |qres|00:02:10|01:41:34|00:20:50|00:01:46|00:24:39|
> |fls|00:01:29|01:38:01|00:13:38|00:01:08|00:11:51|
> |Cheby1KAN|00:12:08|03:32:10|00:50:45|00:03:21|04:24:59|
> |Cheby2KAN|01:06:54|05:03:18|01:35:40|00:03:27|05:41:42|
> |AC-PKAN|00:15:16|01:13:01|01:13:11|00:01:04|02:21:40|
> |KINN|03:04:19|25:00:20|01:51:44|00:14:07|14:31:42|
> |rKAN|01:21:25|12:44:16|06:21:00|00:16:06|05:19:04|
> |FastKAN|05:51:21|09:35:51|03:37:57|00:17:23|02:04:42|
> |fKAN|00:13:09|08:20:34|00:52:05|00:06:22|03:01:41|
> |FourierKAN|01:21:50|03:33:46|07:40:43|00:18:26|02:48:50|
>
> We trained the models until convergence but did not exceed 50,000 epochs. As shown, AC-PKAN still has certain advantages among the KAN model variants, although the running times of all the KAN variants are relatively long. This is primarily because the KAN model is relatively new and in its primitive stages; although it is theoretically novel, its engineering implementation remains coarse and lacks deeper refinements. Moreover, while traditional neural networks benefit from relatively mature optimizers such as Adam and L-BFGS, optimization schemes specifically tailored for KAN have not yet been explored. We believe the performance of AC-PKAN will be further enhanced as the overall optimization performance of KAN variants improves.
>
> **Response to W2**
>
> Please refer to **Ablation Study 2**
>
> **Response to W3**
>
> We reframe the statement of Proposition 1 as:
>
> *Let $\mathcal{N}$ be an AC-PKAN model with $L$ layers ($L \geq 2$) and infinite width. Then, the output $y = \mathcal{N}(x)$ has non-zero derivatives of any finite-order with respect to the input $x$.*
>
> **Response to W4 & Q2**
>
> Please refer to **2D Navier–Stokes PDE Experiment**. Our AC-PKAN model continues to demonstrate strong performance.
>
> **Response to W5**
>
> Please refer to **Ablation Study 1**. Rather than increasing the scaling factor, we consider it more reasonable to multiply the IC/BC loss term ,with its large coefficient $\lambda^{\text{GRA}}$, by a very small scaling factor. This approach ensures that the overall product of IC/BC loss terms’ coefficients approximates the magnitude of overall coefficient of the PDE residual loss term whose scaling factor is set to 1.
>
> The motivation for the logarithmic transformation originates from the Bode plot in control engineering, which uses a logarithmic frequency axis while directly labeling actual frequency values. This technique not only compresses a wide frequency range but also linearizes the system's gain and phase characteristics on a logarithmic scale, thereby alleviating imbalances caused by significant differences in data scales.
>
> **Response to Q1**
>
> In fact, all our experiments use the AdamW optimizer and we clearly stated in the appendix. Step 7 in Algorithm 1 refers to a generic model parameter update step and does not mandate the use of SGD. The focus of our algorithm lies in loss weighting rather than parameter updates. Actually, it is common practice in papers to use gradient descent in pseudo-code as a simplified representation of the parameter update step without specifying a particular optimizer.
>
> **Response to Q3**
>
> It is true that an MLP with sinusoidal activations guarantees non-zero derivatives in its output. However, this advantage specifically addresses improvements to KAN variants rather than MLP-based models. The fitting capability of KAN models relies on polynomial functions with learnable parameters. To ensure non-zero derivatives in the output, the original B-spline KAN includes an additional nonlinear bias function $b(x)$. In contrast, some other KAN variants, such as Cheby1KAN, rely solely on polynomial bases, which inevitably result in zero derivatives when the order of differentiation exceeds the polynomial degree. Furthermore, Proposition 1 was primarily proposed to provide a theoretical guarantee for AC-PKAN's ability to solve any finite-order PDE, analogous to how the universal approximation theorem theoretically establishes the universal fitting capability of neural networks.
>
> **Response to Q4**
>
> Other PINN methods do not require similar linear projections because they are inherently MLP-based models. In AC-PKAN, the linear layer is designed to achieve a hybridization of KAN and MLP architectures. Its role as an initial projection was inspired by the Spatio-Temporal Mixer linear layer in the PINNsformer model, which enhances spatiotemporal aggregation.

---

> > ### Comment · Reviewer_mYon · 2024-11-27
> >
> > Thank you for the detailed explanations and ablation study for W2 & W5. The explanations have resolved my previous questions Q1-Q4. Regarding W4, I think it is better to showcase how AC-PKAN performs beyond a simple 2D Navier–Stokes PDE Experiment and demonstrate its effectiveness in high-dimensional data.
> >
> > For Ablation Study 2: Effect of RGA in Other PINN Variants, the authors did not discuss whether First-kind Chebyshev KAN Layer alone could improve PDE solving ability based only on the PINN objective. The table should also reflect on how RGA improves over Naiive PINN, FourierKAN, etc.
> >
> > I think the authors have made solid contributions and therefore I will increase my score to 6.

---

> > > ### Author Response · Authors · 2024-12-02
> > >
> > > Thank you for your sincere feedback and recognition of our work. Applying AC-PKAN to more specific higher-dimensional experiments is an interesting direction for our future research. Regarding Ablation Study 2, comparing Table 7 in Appendix C2 (the results table of ablation study 2) with Table 2 in the main text allows us to reflect on how RGA improves over vanilla PINN, FourierKAN, and others. By analyzing the comparisons of AC-PKAN, Cheby1KAN, and PINN (including their vanilla versions plus RGA results) in Table 7 and Table 2, we can conclude that the First-kind Chebyshev KAN Layer enhances PDE-solving capabilities.
> > >
> > > However, to enhance the readability of the paper and facilitate the reader's understanding, we promise to add the performance of the Naive PINN variants in Table 7 in the camera-ready version (despite their previous inclusion in Table 2 of the main text). Thank you again for your suggestions!

---

> ### Author Response · Authors · 2024-11-24
>
> Dear Reviewer mYon,
>
> We appreciate your time for reviewing, and we really want to have a further discussion with you to see if our response solves the concerns. We have addressed all the thoughtful questions raised by the reviewer (eg, technical descriptions and contributions), and we hope that our work’s impact and results are better highlighted with our responses. It would be great if the reviewer can kindly check our responses and provide feedback with further questions/concerns (if any). We would be more than happy to address them.
>
> Thank you!
>
> Best wishes,
>
> Authors

---

### Official Review · Reviewer_Cr7S · 2024-11-04

**Soundness:** 3
**Presentation:** 3
**Contribution:** 3
**Rating:** 8
**Confidence:** 3

**Summary:**

This paper proposes a new architecture, the AC-PKAN, which improves on previous variation of Kolmogorov-Arnold Neural Networks thanks to its replacement of the traditional B-spline layers with the novel Chebyshev type-1 polynomial layers. It also introduces an external mechanism, the RGA (residual and gradient based attention), which allows an adaptive reweighting of the various loss components considered and ensures an efficient optimization. The performance of the model is evaluated for various function and PDE solution approximation cases, and an ablation study is proposed.

**Strengths:**

- This contribution innovates by proposing a novel architecture based on the Chebyshev type 1 polynomials to replace the traditional B-splines, as well as an adaptive reweighting technique to control the physics loss
- The newly proposed layer has non-zero derivatives of all orders, a property backed  by a theoretical proof
- The benchmarks show that AC-PKAN offers state-of-the-art (or close to) performance on all cases studied, which include complex PDEs and non-trivial domains
- An ablation study showcases the importance of all the modules introduced in this new architecture, including the RGA

**Weaknesses:**

While the claims are solidly backed by theory and numerical results, the paper sometimes lacks descriptions to aid reader comprehension.
The contribution could provide more explanations for the reason why certain architectural choices are utilized. The role of  the wavelet activation function is one of them.
The paper could be more clear and provide at least some intuition of the explanation for why the Chebyshev type 1 polynomials lead to an increased performance and lower memory cost in comparison to the B-spline.

**Questions:**

Could you provide a more in depth explanation (even intuitive) about (see weaknesses):
- the role of the wavelet activation function and how it helps with performance
- the reason why Chebyshev type 1 polynomials improve upon B-splines

---

> ### Author Response · Authors · 2024-11-18
> **Rebuttal to Reviewer Cr7S**
>
> We thank the reviewer for the thoughtful comments and appreciation for our work.
>
> **Response to Q1**
>
> The wavelet activation function, introduced in [A], is defined as $ \text{Waveact}(x) = \omega_1 \sin(x) + \omega_2 \cos(x) $, with learnable parameters $ \omega_1 $ and $ \omega_2 $. This function combines the benefits of the Fourier feature embedding method from [B] and the sine activation function from [C].
>
> In [B], input coordinates $ x $ are mapped to high-frequency signals as:
> $$
> \gamma(x) = \begin{bmatrix} \cos(Bx) \\ \sin(Bx) \end{bmatrix},
> $$
> where $ B \in \mathbb{R}^{m \times d} $ is a matrix with elements sampled from a Gaussian distribution $ N(0, \sigma^2) $, and $ \sigma $ controls the frequency range.
>
> In [C], the First-Layer Sine (FLS) method applies $ \sin(x) $ as the activation function in the first layer, improving gradient distribution in early training.
>
> The wavelet activation function, employed in the initial encoders $U$ and $V$, retains the advantages of FLS for proper gradient initialization. Intuitively, compared to the pure sine activation function, the wavelet activation introduces additional phase and magnitude information. The arbitrary and learnable combination of phase and magnitude offers greater representational capacity than the fixed phase and magnitude of the pure sine activation function. Moreover, through learnable parameters, it enables adaptive Fourier feature embedding, enhancing the network's ability to capture periodic features and reduce spectral bias.
>
> [A] Zhao Z, Ding X, Prakash B A. Pinnsformer: A transformer-based framework for physics-informed neural networks. arXiv preprint arXiv:2307.11833, 2023.
> [B] Wang S, Sankaran S, Wang H, et al. An expert's guide to training physics-informed neural networks. arXiv preprint arXiv:2308.08468, 2023.
> [C] Wong J C, Ooi C C, Gupta A, et al. Learning in sinusoidal spaces with physics-informed neural networks. IEEE Transactions on Artificial Intelligence, 2022, 5(3): 985-1000.
>
> **Response to Q2**
>
> *1. Performance of Chebyshev Type 1 Polynomials in High-Frequency Scenarios*
>
> Chebyshev polynomials of the first kind, $ T_n(x) = \cos(n \arccos(x)) $, focus their spectrum on high frequencies, with the frequency increasing linearly with polynomial order $ n $. This makes them suitable for capturing high-frequency oscillations, as their high-frequency components decay slowly. The even distribution of extrema further aids in capturing rapid variations, which is beneficial for high-frequency features.
>
> B-splines, being piecewise polynomials, have a rapidly decaying spectrum, limiting their ability to capture high-frequency features.
>
> *2. Global Support and Orthogonality of Chebyshev Polynomials vs. Local Support of B-Splines*
>
> Chebyshev polynomials have both global support and global orthogonality over the interval $[-1, 1]$. The value of a Chebyshev polynomial at any point depends on all points within the interval, making them highly effective at capturing global features and high-frequency components. They satisfy the orthogonality relation:
>
> $$
> \int_{-1}^1 \frac{T_m(x) \\, T_n(x)}{\sqrt{1 - x^2}} \\, dx = 0,\quad \text{for } m \ne n.
> $$
>
> This orthogonality allows Chebyshev polynomials to achieve minimax approximation, minimizing the maximum error over the interval.
>
> In contrast, B-splines have local support; each basis function is nonzero only within a specific subinterval. This local nature limits their ability to capture global high-frequency features. Additionally, B-splines lack global orthogonality, reducing their efficiency in approximating functions.
>
> *3. Memory Efficiency of Chebyshev Polynomials Compared to B-Splines*
>
> B-spline-based KANs require substantial memory due to the storage of grids and coefficient matrices that scale cubically with grid size and spline order. They store grids of size $ (in\\_features , grid\\_size + 2 \times spline\\_order + 1) $ and coefficient matrices of size $ (out\\_features , in\\_features , grid\\_size + spline\\_order) $. Since B-splines are piecewise polynomials, each segment requires maintaining basis function values and performing high-order interpolation within its support interval. This involves generating polynomial bases, solving linear systems (e.g., using `torch.linalg.lstsq`), and executing recursive updates, resulting in high computational and storage demands.
>
> In contrast, Chebyshev polynomials are globally defined and require only a coefficient matrix of size $(input\\_dim , output\\_dim , degree + 1)$, eliminating terms directly related to grid size and spline order. The memory complexity grows linearly with the degree. Chebyshev polynomials eliminate the need for grid storage and do not require solving linear systems, interpolation, or recursive updates of piecewise basis functions, which significantly reduce computational and storage requirements.

---

> > ### Comment · Reviewer_Cr7S · 2024-11-23
> >
> > Thank you for the reply. These detailed explanations greatly improve the clarity of the method and improve the reader's understanding for the advantages brought by the use of wavelet activations and the selection of expansions in Chebyshev type 1 polynomials. I maintain my rating for this contribution.

---

### Author Response · Authors · 2024-11-18
**Rebuttal to All Reviewers (Common Response)**

In response to several reviewers' requests for some ablation studies and more challenging PDE experiment, we are pleased to present the following three experiments:

**2D Navier–Stokes PDE Experiment**

The 2D Navier–Stokes equations are given by:

$$
\begin{aligned}
    \frac{\partial u}{\partial t} + \lambda_1 \left(u \frac{\partial u}{\partial x} + v \frac{\partial u}{\partial y}\right) &= -\frac{\partial p}{\partial x} + \lambda_2 \left(\frac{\partial^2 u}{\partial x^2} + \frac{\partial^2 u}{\partial y^2}\right), \\\\
    \frac{\partial v}{\partial t} + \lambda_1 \left(u \frac{\partial v}{\partial x} + v \frac{\partial v}{\partial y}\right) &= -\frac{\partial p}{\partial y} + \lambda_2 \left(\frac{\partial^2 v}{\partial x^2} + \frac{\partial^2 v}{\partial y^2}\right),
\end{aligned}
$$

where $ u(t, x, y) $ and $ v(t, x, y)$ are the $ x $-component and $y$-component of the velocity field, respectively, and $ p(t, x, y) $ is the pressure. This equation describes the Navier–Stokes flow around a cylinder. All the experiment seetings are same with [A]. The following table shows the test errors for $ v $:

|Model|rMAE|rRMSE|
|-|-|-|
|PINN|3.6949|3.2899|
|PINNsformer|3.6986|3.2924|
|qres|3.2930|3.6998|
|fls|3.6930|3.2893|
|Cheby2KAN|**3.0443**|2.9513|
|AC-PKAN|***2.4519***|**2.4412**|
|KINN|3.6816|3.2801|
|FastKAN|3.6999|***1.3401***|
|fKAN|3.7040|3.2998|
|FourierKAN|5672.3763|5973.1545|
|Cheby1KAN|3.7561|3.3347|
|rKAN|NaN|NaN|

[A]  Raissi, Maziar, Paris Perdikaris, and George E. Karniadakis. "Physics-informed neural networks: A deep learning framework for solving forward and inverse problems involving nonlinear partial differential equations." Journal of Computational physics 378 (2019): 686-707.

 **Ablation Study 1: Effect of Log Transformation**

We removed the log transformation in RGA module across five PDE experimental tasks and set the scaling factors to smaller numbers. Specifically, we used the original RGA design to pre-train for several epochs, obtaining very large values of $ \lambda^{\text{GRA}} $. To ensure consistency in the magnitudes of different loss terms, we set the scaling factor of the PDE-residual loss term to 1 and assigned the scaling factors of the data loss terms (including BC/IC) to the negative order of magnitude of the current $\lambda^{\text{GRA}}$.

The results are summarized in the following table:

|Equation|rMAE|rRMSE|
|-|-|-|
|1D Wave|0.7686|0.7479|
|1D Reaction|2.2348|2.2410|
|Heterogeneous Problem|10.0849|9.6492|
|Complex Geometry|164.4283|158.7840|
|Navier–Stokes|84.3943|88.7684|

After removing the log transformation, AC-PKAN's performance significantly deteriorated compared to Tables 2 and 3 in the manuscript. This decline occurs because $\lambda^{\text{GRA}}$ is not only excessively large in data scale but also exhibits a wide range of variation. During the original training process, the coefficient $\lambda^{\text{GRA}}$ rapidly grows from 0 to a very large value, displaying a broad dynamic range. The logarithmic transformation narrows this range substantially; for example, in the 1D Wave experiment, the scale of $\lambda$ versus epochs ranges from 0 to $4 \times 10^7$, whereas $\ln(\lambda)$ ranges from 7 to 15 graphs will be attached in the appendix of the revised manuscript). Removing the log transformation and forcibly setting scaling factors at similar apparent magnitudes is ineffective. The model cannot adapt to the drastic changes in $\lambda^{\text{GRA}}$, and rigid manual scaling factors exacerbate the imbalance among loss terms, ultimately leading to training failure. Through confining the variation range of $\lambda^{\text{GRA}}$, the log transformation allows the model to adjust more flexibly and effectively.

**Ablation Study 2: Effect of RGA in Other PINN Variants**

We adapted our RGA module to other algorithms, and the experimental results are as follows:

|Model|rMAE|rRMSE|
|-|-|-|
|PINN+RGA|0.0914|0.0924|
|PINNsformer+RGA|NaN|NaN|
|qres+RGA|0.2204|0.2184|
|fls+RGA|0.1610|0.1617|
|Cheby1KAN+RGA|0.0567|0.0586|
|Cheby2KAN+RGA|1.0114|1.0048|
|AC-PKAN|***0.0011***|***0.0011***|
|KINN+RGA|**0.0479**|**0.0486**|
|rKAN+RGA|NaN|NaN|
|FastKAN+RGA|0.1348|0.1376|
|fKAN+RGA|0.2177|0.2149|
|FourierKAN+RGA|1.0015|1.0001|

In the case of PINNsformer+RGA, the result is NaN due to a CUDA out-of-memory error during training. This occurs because PINNsformer needs to create pseudo sequences. And applying RGA, which requires gradient computation on a large number of data points within the pseudo sequence, incurs significant memory overhead, leading to training failure. Meanwhile, rKAN+RGA resulted in NaN due to gradient instability during training.

Except for these cases, compared to Tables 2 and 3 in the paper, the performance of other models significantly improved, indicating that our RGA can be generally transferred to other models to enhance performance. However, their performance still does not surpass that of our AC-PKAN.

---

### Meta-Review · Area_Chair_RDEr · 2024-12-22

**Metareview:**

The paper proposes a new variant of Kolmogorov-Arnold Network (KAN) and also a new learning scheme with 'gradient-based attention' and 'residual-based attention'. The resulting model in the provided experiments achieves best or second-best results (see Table 2 in the text). Overall, the idea of using additional components to the loss functions to improve generalization of KAN (or MLP, or other deep learning models) is always promising.

However, there are concerns about the actual effectiveness of the method and the comparison. For example, as pointed out by one of the reviewers, the PINNsFormer model allows to reach much better effectiveness, and the authors have answered, that it is due to 'external learning strategies'. However, there own method has at least two components of the loss that do not look intuitive. Thus, the question wether there is a real improvement over state of art is unclear.

Moreover, the example considered are not challenging.

My main concern is that the method contains a lot of 'random' tricks (wavelet MLP, Chebyshev KAN, 'residual and gradient based attentions' all of which are different contributions, which do not fit into a single pape

**Additional Comments On Reviewer Discussion:**

The discussion was split: some of the reviewers are highly favorable of the paper, but some were more skeptical. The authors tried to address the questions with additional experiments and comments. I think reviewer w3dV evaluation is much more close to mine, especially in terms of how challenging are considered PDEs and how accurate is the comparison with previous work.

---

### Decision · Program_Chairs · 2025-01-22

Reject